# AIMS: All-Inclusive Multi-Level Segmentation for Anything

**Lu Qi**[1]    **Jason Kuen**[2]    **Weidong Guo**[3*]    **Jiuxiang Gu**[2]
**Zhe Lin**[2]    **Bo Du**[4]    **Yu Xu**[3]    **Ming-Hsuan Yang**[1,5]
[1]UC Merced    [2]Adobe Research    [3]QQ Brower Lab, Tencent
[4] Wuhan University    [5] Google Research

## Abstract

Despite the progress of image segmentation for accurate visual entity segmentation, completing the diverse requirements of image editing applications for different-level region-of-interest selections remains unsolved. In this paper, we propose a new task, All-Inclusive Multi-Level Segmentation (AIMS), which segments visual regions into three levels: part, entity, and relation (two entities with some semantic relationships). We also build a unified AIMS model through multi-dataset multi-task training to address the two major challenges of annotation inconsistency and task correlation. Specifically, we propose task complementarity, association, and prompt mask encoder for three-level predictions. Extensive experiments demonstrate the effectiveness and generalization capacity of our method compared to other state-of-the-art methods on a single dataset or the concurrent work on segment anything. We will make our code and training model publicly available.

## 1 Introduction

Decomposing an image into semantically meaningful regions is a key goal of image segmentation. With the recent emergence of photorealistic and high-quality image generation technologies [1, 2], image segmentation has gained considerable attention for its ability to select visual entities of interest within images. Combining image segmentation with image generation has enabled a variety of image editing and manipulation applications, such as altering the style of a specific object or eliminating a distracting element from a scene. Compared to bounding box-based object detection, segmentation methods preserve the integrity of pixels outside the edited or manipulated region.

Panoptic segmentation [3], with its instance awareness and dense segmentation capabilities, is a promising solution for various editing and manipulation applications. However, two main limitations remain when using it: (1) it struggles to handle classes encountered 'in the wild' but absent in the training set [4, 5]; and (2) existing methods typically do not explore the more general possibilities of segmentation masks across multiple abstraction levels (*e.g., a car's wheel*, *the whole car*, both *a person and car* combined), with the exception of part-aware panoptic segmentation [6, 7] and panoptic scene graph [8]. These tasks also fail to generalize to classes beyond those in the training set, especially at the part-level task where annotations are sparse.

In this paper, we address the constraints inherent to different-level segmentation tasks by introducing **AIMS** (All-Inclusive Multi-Level Segmentation). AIMS represents a comprehensive framework designed to segment images across various hierarchical levels (*part*, *entity*, *relation*) using multiple query-driven image decoders. Drawing inspiration from recent advancements in Entity Segmentation [4, 5], which showcased remarkable generalization capabilities on unseen classes, AIMS performs class-agnostic segmentation across different hierarchical levels to enhance generalization.

---

*Corresponding author.

37th Conference on Neural Information Processing Systems (NeurIPS 2023).

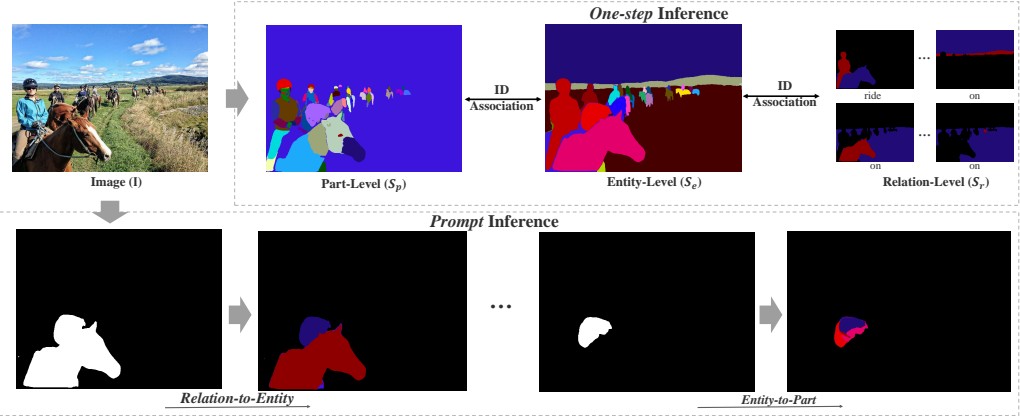

Figure 1: The overview of our All-Inclusive Multi-Level Segmentation (AIMS) task.

Thanks to its hierarchical modeling and class-agnostic learning approach, AIMS provides more level-flexible and all-inclusive segmentation masks for users to select from, thereby enabling a wider range of image editing and manipulation applications. In Figure 1, the AIMS task has two modes of inference: one-step inference and prompt inference. In one-step inference, the model segments the image directly into three levels (part, entity, and relation level). Meanwhile, in prompt inference, the model separates the image further based on the provided mask. Notably, one-step inference necessitates an association of the results between any two neighboring levels.

Unlike existing part-aware panoptic or panoptic scene graph methods, AIMS model is not confined to training on a single dataset that has limited hierarchical levels. Instead, we adopt the strategy of multi-dataset training to build a unified AIMS model for all levels. To maintain consistency across multiple hierarchical levels sourced from different datasets, we introduce the *Task Complementarity Module* and the *Association Module* to AIMS model. These respectively facilitate feature interactions between different levels and establish explicit relationships between predictions at adjacent levels (*e.g.,* a car's part mask should correspond with the overall car mask, and a car mask should belong to a mask with a car and driver.).

We train our AIMS model on existing segmentation datasets such as Pascal Panoptic Parts [6], COCO-PSG [8], PACO [9], and EntitySeg [5]. However, due to partial or incomplete annotations over the full image, with inconsistencies in unannotated areas across different datasets, sub-optimal supervision signals are generated during training. To address this issue, we propose a novel *Mask Prompt Encoder* for our AIMS model. This encoder segments an image based on a given mask prompt, which specifies the region slated for segmentation. We train AIMS to accommodate a diverse set of mask prompts, derived from various hierarchical levels and datasets while excluding unannotated regions. This encoder offers two distinct benefits: *(1)* it mitigates the inconsistency problem inherent in training with partial annotations from multiple datasets; *(2)* it exhibits strong generalizability to arbitrary mask prompts during inference, including those stemming from unseen classes and in-the-wild visual entities, as shown in Figure 5.

The main contributions of this work are as follows:

- We introduce the AIMS task, which segments images into three distinct levels: part, entity, and relation. AIMS is capable of performing a one-step inference for the entire image and a prompt inference for a specified mask area within the image, which has great potential for image editing and manipulation applications.

- We propose a unified AIMS model for this task, which incorporates multi-dataset and multi-task training. We have meticulously designed several modules, including the task complementary, association, and mask prompt encoder, to maintain the consistency of tasks or data annotations.

- Extensive experiments and in-depth analysis showcase the effectiveness of our proposed AIMS model across various segmentation tasks and settings. We compare the results from our AIMS model with the popular segmentation models (*e.g.*, SAM[10]), demonstrating the fine-grained segmentation capabilities and generalization to both seen and unseen categories.

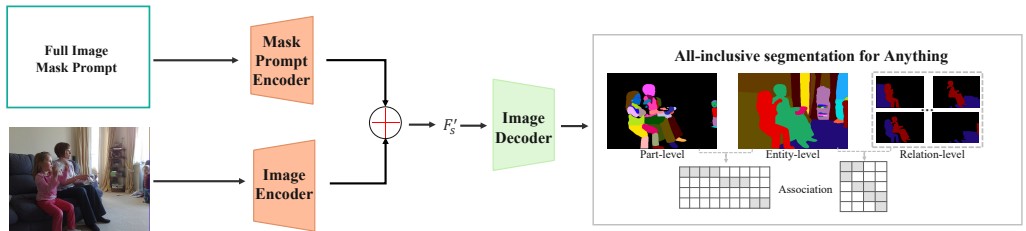

Figure 2: The outline of our AIMS framework encompasses an image encoder, a mask prompt encoder, and an image decoder. The outputs generated by AIMS consist of five distinct components, including segmentation results across three levels, and two association metrics.

## 2 Related Work

**Image Segmentation** Significant advancements [11, 12, 13, 14, 15, 16, 17, 18, 19, 20, 21] have been made in image segmentation [22, 23, 3, 24, 25, 26] in recent years, with numerous methods and techniques emerging to handle various challenges through supervised learning. Notable works include FCN [11], DeepLab [12], PSPNet [13], Mask R-CNN [14], PANet [15], SOLO [16], CondInst [17], PanopticFPN [18], PanopticFCN [19], Max-Deeplab [20], and Mask2Former [21] for semantic, instance, and panoptic segmentation. Additionally, specialized tasks tailored to specific scenarios, such as part-aware panoptic segmentation [6, 27, 7, 28] and panoptic scene graphs [8, 29], have been developed. However, these tasks typically focus on single target and do not incorporate hierarchical concepts that are more in line with human behavior. This paper aims to address this problem by proposing a unified segmentation model that can organize pixels at three different levels: part, entity, and relation. We also introduce an interactive stage to decompose unseen regions into smaller entities or parts in an open-world setting.

**Multi-Dataset Training** Multi-dataset training has gained considerable attention in recent years, as it allows models to draw from multiple data sources to improve performance and generalization [30, 31, 32]. This approach has significantly benefited domain adaptation [33, 34, 35], dataset distillation [36, 37], curriculum learning [38, 39, 40, 41], data augmentation [42, 43, 44], federated learning [45, 46], and transfer learning [47, 48] in the context of image segmentation. This paper addresses the annotation inconsistency among multiple datasets for generalized hierarchical segmentation. Specifically, we tackle the issue of incomplete annotations in a single dataset across three levels using a prompt structure.

## 3 Method

As shown in Figure 1, given an image $\mathbf{I} \in \mathbb{R}^{H \times W \times 3}$, the AIMS task segments an image into three segmentation maps $\mathbf{S}_T \in \{0, 1\}^{N_T \times H \times W}$ at part, entity, and relation levels and two association maps $\mathbf{A_{EP}} \in \{0, 1\}^{N_e \times N_p}$ (entity and part levels) and $\mathbf{A_{RE}} \in \{0, 1\}^{N_r \times N_e}$ (relation and entity levels), where $H$ and $W$ are the height and width of the image, $N_T$ indicates $N_T$ prediction results in class-agnostic binary format and $T = \{\mathbf{p}, \mathbf{e}, \mathbf{r}\}$. The two association maps $\mathbf{A_{EP}}$ and $\mathbf{A_{RE}}$ are required to build the association relationships between two level prediction results. We chose these three levels since they align with the human brain's hierarchical information processing system [49, 50, 51] and fulfill the majority of requirements for image editing tools [52]. The outputs of AIMS are akin to those of related tasks like part-aware panoptic segmentation and panoptic scene graphs, but the AIMS task incorporates more levels and operates in a class-agnostic manner.

Like the existing Transformer-based segmentation methods [21], our AIMS model employs an image encoder and decoder to extract robust image features and decode them into segmentation masks. However, it differs in that it addresses the challenges of multi-dataset multi-task training. As illustrated in Figure 3, we first establish a task complementarity module and an association module to build the correlation of each task, followed by a mask prompt encoder to tackle the partial annotation problem among different datasets and enable generalized inference.

Next, we introduce our base framework and then elaborate on our three proposed components.

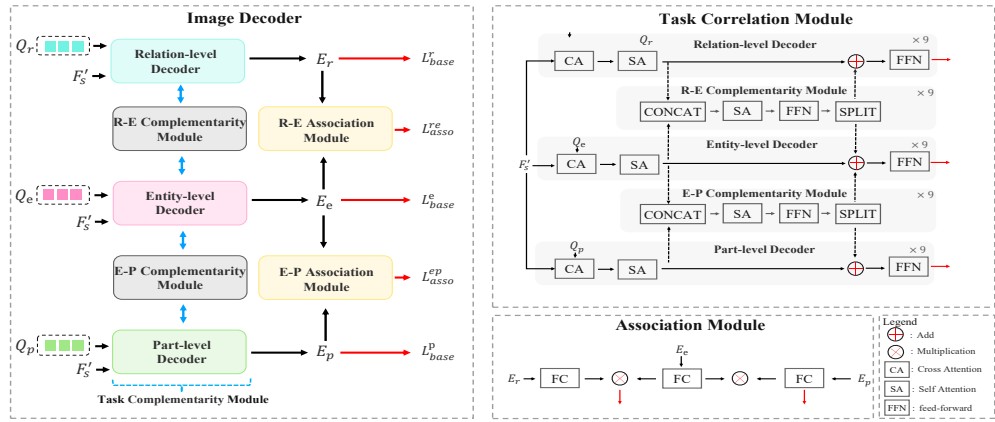

Figure 3: Overview of image decoder of our All-Inclusive Multi-Level Segmentation framework. The terms *'E-P Correlation Module'* and *'E-P Association Module'* refer to the task correlation and association structure between the entity and part level, respectively. This explanation can also be applied to the *'R-E Correlation Module'* and *'R-E Association Module'*. The rightward arrows denote the outputs from the respective modules. *'CONCAT'* or *'SPLIT'* refers to the joining of two tensors or the division of a single tensor into two tensors, respectively, occurring along the dimension of the query number.

## 3.1 Base Framework

The base framework of AIMS shares the same image encoder $\Theta_I$ and has three independent decoders for different-level predictions. The image encoder consists of a backbone and pixel decoder, extracting multi-scale image features $\mathbf{F_s} \in \mathbb{R}^{\frac{H}{2^s} \times \frac{W}{2^s} \times 256}$ and $s = \{2,3,4,5\}$. We denote this image-encoding process as:

$$\mathbf{F_s} = \Theta_I(\mathbf{I}) \tag{1}$$

For the three independent decoders $\Phi$, each decoder has 9 cascaded attention blocks, where each block includes a masked cross-attention, self-attention, and feedforward network. Each block of decoder iteratively refines query embeddings $\mathbf{E_T} \in \mathbb{R}^{N \times 256}$ by interacting with multi-scale features $F$, where $\mathbf{E_T}$ is initialized by a group of learnable parameters $\mathbf{Q_T}$. This iterative decoding process is denoted as

$$\mathbf{E_T} = \Phi_{\mathbf{T}}(\mathbf{Q_T}, \mathbf{F_s}) = \prod_{i=1}^{9} \Phi_{\mathbf{T}}^i(\mathbf{E_T^i}, \mathbf{F_s}) = \prod_{i=1}^{9} R_{\mathrm{ff}_{\mathbf{T}}}^i(R_{\mathrm{SAtt}_{\mathbf{T}}}^i(R_{\mathrm{XAtt}_{\mathbf{T}}}^i(\mathbf{E_T^i}, \mathbf{F_s}))) \tag{2}$$

where $\Phi_{\mathbf{T}}$ and $\Phi_{\mathbf{T}}^i$ indicate the decoder of $\mathbf{T}$ type and its attention blocks. $i$ indicates the $i^{\mathrm{th}}$ attention block. For each attention block, it includes cross-attention $R_{\mathrm{XAtt}_{\mathbf{T}}}^i$, self-attention $R_{\mathrm{SAtt}_{\mathbf{T}}}^i$, and feedforward network $R_{\mathrm{ff}_{\mathbf{T}}}^i$. At last, $\mathbf{E_T}$ is used for part-ness/entity-ness/relation-ness prediction and pixel-level mask prediction using the low-level image feature $\mathbf{F_2}$ (derived from the image encoder). The prediction process can be formulated as:

$$\mathbf{U_T^c}, \mathbf{U_T^m} = \mathrm{PredHead}_{\mathbf{T}}(\mathbf{E_T}, \mathbf{F_2}) \tag{3}$$

where $\mathbf{U_T^c}$ and $\mathbf{U_T^m}$ denote the part-ness/entity-ness/relation-ness prediction and pixel-level mask outputs.

During training, we employ the losses $\mathcal{L}_{\mathbf{T}}$ for different level predictions, with respect to the corresponding ground truth $\mathbf{G_T}$. The overall training loss for our model is defined as:

$$\mathcal{L}_{\mathrm{base}} = \sum_{\mathbf{T} \in \{\mathbf{p}, \mathbf{e}, \mathbf{r}\}} \mathcal{L}_{\mathbf{T}}^{\mathrm{ce}}(\mathbf{U_T^e}, \mathbf{G_T^e}) + \mathcal{L}_{\mathbf{T}}^{\mathrm{bce}}(\mathbf{U_T^m}, \mathbf{G_T^m}) + \mathcal{L}_{\mathbf{T}}^{\mathrm{dice}}(\mathbf{U_T^m}, \mathbf{G_T^m}) \tag{4}$$

where $\mathcal{L}_{\mathbf{T}}^{\mathrm{ce}}$ denote binary cross-entropy loss for part-/entity-/relation-ness prediction. Similarly, $\mathcal{L}_{\mathbf{T}}^{\mathrm{bce}}$ and $\mathcal{L}_{\mathbf{T}}^{\mathrm{dice}}$ denote the binary cross-entropy and dice loss for different level mask prediction.

| Level | COCO | PPP | COCO-PSG | PACO | EntitySeg |
|---|---|---|---|---|---|
| Relation | ○ | ○ | ✓ | ○ | ○ |
| Entity | ✓ | ✓ | ✓ | ✓ | ✓ |
| Part | ○ | ✓ | ○ | ✓ | ○ |

(a)

| Information | COCO | PPP | COCO-PSG | PACO | EntitySeg |
|---|---|---|---|---|---|
| Annotation Area | 89.1% | 91.7% | 90.4% | 8.4% | 99.9% |
| Category Numbers | 133 | 20 | 133 | 75 | 634 |
| Image Numbers | 115k | 9.7k | 46k | 35k | 32k |

(b)

Table 1: The statistics of the datasets we used for training including COCO [53], EntitySeg [5], PascalVOC Part (PPP) [6], PACO [9], and COCO-PSG [8]. The (a) indicates whether the dataset has the annotations for some level. The (b) statisfy the information of each dataset.

## 3.2 Task Complementarity Module

In our base method, we utilize three independent decoders for predictions at different hierarchy levels, which merely share the image encoder. To better exploit cross-task knowledge, we design the task complementarity module (TCM) that fuses task information within each attention block of the decoders. To provide a clearer explanation, we use the task complementarity module between the entity and part tasks as an example, and it works identically for the relation and entity tasks. As depicted in Figure 3, we concatenate the embeddings of two tasks following the self-attention of the attention block between two level decoders. Subsequently, we feed the concatenated embedding into a self-attention $G_{\text{SAtt}}^i$ and feed-forward network $G_{\text{SAtt}}^i$ and then separate processed embeddings back into their original dimensions:

$$\mathbf{E_{e_{s'}}^i}, \mathbf{E_{p_{s'}}^i} = G_{\text{split}}^i(G_{\text{ff}}^i(G_{\text{SAtt}}^i(G_{\text{Concat}}^i(\mathbf{E_{e_s}^i}, \mathbf{E_{p_s}^i})))) \tag{5}$$

where the $\mathbf{E_{e_s}^i}$ and $\mathbf{E_{p_s}^i}$ denote the embedding processed by self-attention in the attention block. After that, we acquire the final embedding by performing a straightforward addition between the fused and original embeddings:

$$\mathbf{E_{e_{s''}}^i} = \mathbf{E_{e_{s'}}^i} + \mathbf{E_{e_s}^i}, \ \ \mathbf{E_{p_{s''}}^i} = \mathbf{E_{p_{s'}}^i} + \mathbf{E_{p_s}^i} \tag{6}$$

Ultimately, we feed the enhanced embeddings at each level into the original feedforward network of the corresponding decoder:

$$\mathbf{E_T^{i+1}} = R_{\text{ff}_T}^i(\mathbf{E_{T_{s''}}^i}). \tag{7}$$

## 3.3 Association Module

Aside from generating predictions at various levels, the AIMS model also strives to establish the association between low-level and higher-level results. For example, the part masks of a car should be explicitly associated with the entire car's mask. In image editing and manipulation, this can provide a more intuitive mask selection experience. While it is possible to associate the masks across different levels by computing mask intersection, such association knowledge cannot be instilled into the model. Instead, we introduce an association module to AIMS to learn cross-level associations by applying an association loss to the similarity between embeddings. Besides having the association graph as an outcome, this also helps the model be aware of the cross-level association relationships and thus benefiting the individual-level task performance. The association loss $\mathcal{L}_{\text{asso}}$ is defined as:

$$\mathcal{L}_{\text{asso}} = \mathcal{L}_{\text{bce}}(\text{FC}(\mathbf{E_e}) \cdot (\text{FC}(\mathbf{E_p}))^T, \mathbf{G_{ep}}) + \mathcal{L}_{\text{bce}}(\text{FC}(\mathbf{E_r}) \cdot (\text{FC}(\mathbf{E_e}))^T, \mathbf{G_{re}}) \tag{8}$$

where FC indicates a *fully-connected* layer and $\cdot$ denotes dot product. The $\mathbf{G_{ep}} \in \{0, 1\}^{N_e \times N_P}$ and $\mathbf{G_{re}} \in \{0, 1\}^{N_r \times N_e}$ are the binary association ground truth of entity-part and relation-entity levels. Finally, the overall training loss for AIMS is

$$\mathcal{L} = \mathcal{L}_{\text{base}} + \mathcal{L}_{\text{asso}} \tag{9}$$

## 3.4 Mask Prompt Encoder

As aforementioned, we train AIMS with multiple existing segmentation datasets that each covers one or more hierarchy levels, whose details are provided in Table 1. Due to the use of predefined classes, most of the datasets suffer from the problem of partial annotations and have unannotated areas on the images. Furthermore, the unannotated areas appear and happen inconsistently among the multiple datasets (*e.g.,* 91.6% of PACO is unannotated, while it is 0.01% for EntitySeg). This leads to undesirable supervision signals that confuse the model because something that is treated as *background* in one dataset/level may qualify as *foreground* in another dataset/level.

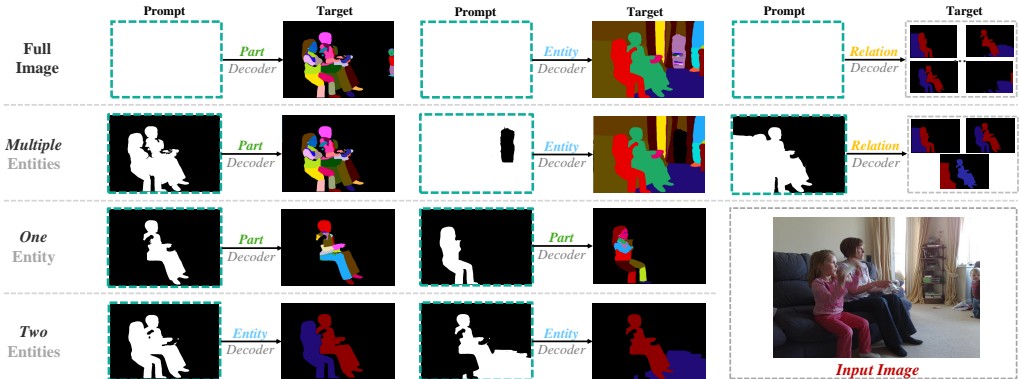

Figure 4: Illustration of the task prompt types with different combinations between mask prompts and decoders. For each sub-figure, the right part is our target of the network.

We design the Mask Prompt Encoder (MPE), which allows our method to segment images conditioned on specific mask-controlled task prompts. As depicted in Figure 4, our approach integrates four types of mask prompts: full image, single entity, dual entities and multiple entities (where the upper limit is the entire annotated area). By employing diverse mask prompts along with multi-level mask decoders, AIMS can accomplish different segmentation goals. The mask prompts, covering either the full or partial image area, drive the model to segment the entire image or target area at part, entity, or relation levels using the corresponding decoders. This approach enables us to provide the model with a trustworthy supervision signal for the region to be segmented, even when dealing with datasets that lack full image annotations. Moreover, during inference, the mask prompt encoder can be applied to unrestricted mask prompts. To some extent, the model can subdivide the masks of unseen classes and/or in-the-wild visual entities into reasonable sub-masks, even if they are absent in training data.

In particular, the mask encoder accepts a binary mask $\mathbf{M_P}$, which designates the area to be segmented. Our mask encoder utilizes a series of cascaded convolution layers to downscale $\mathbf{M_P}$, resulting in $\mathbf{M_2} \in \mathbb{R}^{\frac{H}{4} \times \frac{W}{4} \times C_m}$.

$$\mathbf{M_2} = \text{MaskEncoder}(\mathbf{M_P}) \tag{10}$$

where the detailed structure of the MaskEncoder consists of three operation blocks including CONV2+LN+GELU, CONV2+LN+GELU and CONV1. The 'CONV2', 'CONV1', 'LN', and 'GELU' represent a convolution layer with stride and a kernel size of 2 or 1, layer normalization, and Gaussian error linear units, respectively. Subsequently, we directly downscale $\mathbf{M_2}$ by $2\times$, $4\times$, and $8\times$ to obtain $\mathbf{M_s}$, where $s \in 3, 4, 5$, using bilinear interpolation.

Ultimately, we acquire multi-scale fusion features $\mathbf{F}'s$ by summing the encoded image and mask features $\mathbf{F}s$ and $\mathbf{M}_s$, respectively. For AIMS with MPE, we replace $\mathbf{F}_s$ in Eq. 2 with $\mathbf{F}'_s$:

$$\mathbf{F}'_s = \mathbf{F}_s + \mathbf{M}_s \tag{11}$$

Given MPE, we delve deeper into the sampling strategy implemented across various mask prompts and subtasks. To effectively balance contributions from each subtask and prompt type, we adopt a uniform sampling approach capable of considering multiple factors, such as subtask difficulty and the distribution of different prompt types within the training data. Specifically, in each training iteration, we have five subtasks: segmenting the entire annotated area (either the full image or multiple entities) based on three hierarchy levels, subdividing a single entity to parts, and subdividing two merged entities with relationships to two independent entities. Additionally, each subtask has its own data pool for sampling, taking into account the statistical annotation information of each dataset. For more details, please refer to our supplementary file.

## 4   Experiments

We construct our training set by aggregating images from five segmentation datasets, including COCO [53], EntitySeg [5], PascalVOC Part (PPP) [6], PACO [9], and COCO-PSG [8]. Given that

| Method | Train Data | Backbone | Inference(PPP) | | | | | | | Inference(COCO-PSG) | | | | | | |
|---|---|---|---|---|---|---|---|---|---|---|---|---|---|---|---|---|
| | | | $AP^P$ | $AP^P_{50}$ | $AP^P_{75}$ | $AP^E$ | $AP^E_{50}$ | $AP^E_{75}$ | $AR^{EP}$ | $AP^R$ | $AP^R_{50}$ | $AP^R_{75}$ | $AP^E$ | $AP^E_{50}$ | $AP^E_{75}$ | $AR^{RE}$ |
| Baseline (Sec. 3.1) | PPP [6] | Swin-Tiny | 24.3 | 54.3 | 18.1 | 48.6 | 70.0 | 49.8 | 65.8 | - | - | - | 25.7 | 53.1 | 34.1 | - |
| | COCO-PSG [8] | | - | - | - | 44.5 | 65.6 | 46.1 | - | 39.6 | 59.7 | 41.3 | 41.7 | 67.3 | 42.4 | 51.8 |
| AIMS | All | | 24.5 | 55.4 | 18.2 | 53.4 | 75.4 | 55.8 | 69.7 | 38.9 | 59.0 | 40.7 | 40.4 | 65.0 | 41.2 | 50.9 |
| | | | 26.5 | 59.0 | 20.2 | 56.1 | 78.5 | 58.5 | 72.3 | 40.5 | 60.6 | 42.3 | 42.1 | 67.2 | 43.3 | 53.1 |
| | | Swin-Large | **29.3** | **63.0** | **23.1** | **60.0** | **81.6** | **63.1** | **75.6** | **44.2** | **65.1** | **45.5** | **46.8** | **70.4** | **48.7** | **56.9** |

Table 2: Comparison among single dataset training, baseline, and our proposed method AIMS on multi-dataset training.

each dataset was originally created for a distinct task, we split each dataset into training and evaluation, and subsequently merge them to form unified training and evaluation sets. Overall, our training and evaluation sets have about 236.7K and 2.1K images. Please refer to our supplementary file for details on the split criteria. With the exception of COCO and EntitySeg, which are solely for entity-level prediction with full or partial image mask prompts, PPP dataset is utilized for entity/part prediction with full or partial image mask prompts and the entity-to-part task. COCO-PSG is employed for relation/entity prediction with full or partial image mask prompts and a relation-to-entities task. We keep only the relationship annotations in COCO-PSG that involve physical contact, to be consistent with typical part-whole relationships. PACO dataset is exclusively used for the entity-to-part task.

**Training settings.** We train our model for 36,000 iterations using a base learning rate of 0.0001 and weights pre-trained on COCO-Entity [3] with the exception of images contained in our validation set. The longer edge size of the images is set to 1,333 pixels, while the shorter edge size is randomly sampled between 640 and 800 pixels, with a stride of 32 pixels. The learning rate is decayed by a factor of 0.1 after 28,000 and 33,000 iterations, respectively. During each training iteration, we sample the data and tasks as introduced in the sampling strategy of Section 3.4, with a batch size of 64 on 8 A100 GPUs.

**Federated evaluation.** We build our evaluation set by collecting images from PPP [6] and COCO-PSG [8] dataset, where each image only has two level annotations. Similar to LVIS [46], our evaluation images have non-exhaustive annotations. To better approach this, we organize the evaluation set into two subsets: one for entity-part and another for relation-entity. For each subset, we evaluate our model by the instance segmentation metric *mean average precision (mAP)* for each level and recall AR@100 for cross-level association predictions. For brief illustration, $AP^p$, $AP^e$, and $AP^r$ are mAP for part-, entity- and relation- predictions. The $AR^{ep}$@100 and $AR^{re}$@100 are metrics for entity-part and relation-entity association performance.

In the following, we first ablate our experiments with a Swin-Tiny [54] backbone and then report the quantitative performance and visualization results of our best model with the Swin-Large [54] backbone to other state-of-the-art methods.

## 4.1 Ablation Study

**Multi-dataset training.** Table 2 demonstrates the overall performance improvement by introducing multiple datasets. In the 1st and 2nd rows, we directly apply the base framework (baseline) with single dataset training and separate decoders. From the inference results, we observe that the model trained on the PPP dataset yields lower performance at the entity level when evaluated on the COCO-PSG dataset. This is primarily due to the fact that the COCO-PSG dataset contains 4 times more images than the PPP dataset, which does not provide sufficient entity-level annotations for the model. This issue can be alleviated through multi-dataset training, as depicted in the third row.

When we train our baseline method with all datasets, there are slight improvements or even degradation in part-level prediction for the PPP dataset, and relation and entity-level prediction for the COCO-PSG dataset. This can be attributed to annotation inconsistencies and incompleteness among multiple datasets, which results in ambiguous training signals for the model. By exploiting task complementarity (with Task Complementarity Module) and mitigating annotation issues (with MPE) , our proposed AIMS framework can improve performance on both evaluation datasets.

**Improvement trajectory of the proposed method.** In Table 3(a), we show the component-wise improvements over the baseline. The 1st and last rows represent the performance of our baseline (three separate decoders for three levels) and the full AIMS framework. The middle three rows display the performance enhancement over the baseline method when integrated with different modules. It is evident that both MPE and TCM contribute to improvements in performance across all three

| MPE | TCM | AM | PPP | | | COCO-PSG | | |
|---|---|---|---|---|---|---|---|---|
| | | | $AP^P$ | $AP^E$ | $AR^{EP}$ | $AP^R$ | $AP^E$ | $AR^{RE}$ |
| ○ | ○ | ○ | 24.5 | 53.4 | 69.7 | 38.9 | 40.4 | 50.9 |
| ✓ | ○ | ○ | 25.7 | 54.8 | 71.0 | 39.5 | 41.2 | 52.0 |
| ○ | ✓ | ○ | 25.9 | 55.0 | 71.2 | 39.9 | 41.6 | 52.3 |
| ○ | ○ | ✓ | 24.3 | 53.1 | 70.6 | 39.0 | 40.5 | 52.0 |
| ✓ | ✓ | ✓ | **26.5** | **56.1** | **72.3** | **40.5** | **42.1** | **53.1** |

(a)

| FUM | PAM | REM | EPM | PPP | | | COCO-PSG | | |
|---|---|---|---|---|---|---|---|---|---|
| | | | | $AP^P$ | $AP^E$ | $AR^{EP}$ | $AP^R$ | $AP^E$ | $AR^{RE}$ |
| ○ | ○ | ○ | ○ | 24.5 | 53.2 | 69.6 | 38.9 | 40.3 | 51.0 |
| ✓ | ○ | ○ | ○ | 24.5 | 53.4 | 69.7 | 38.9 | 40.4 | 50.9 |
| ○ | ✓ | ○ | ○ | 22.1 | 51.8 | 65.8 | 35.8 | 37.7 | 47.6 |
| ✓ | ✓ | ○ | ○ | 25.6 | 54.9 | 70.4 | 39.6 | 41.3 | 52.1 |
| ○ | ○ | ✓ | ✓ | 20.4 | 49.7 | 62.3 | 32.5 | 35.6 | 44.8 |
| ✓ | ✓ | ✓ | ✓ | **26.5** | **56.1** | **72.3** | **40.5** | **42.1** | **53.1** |

(b)

Table 3: Ablation studies. **(a)** Improvement trajectory of the proposed modules. The abbreviations 'MPE', 'TCM', and 'AM' represent the mask prompt encoder, task complementarity module, and association module, respectively. **(b)** Ablation study concerning the utilization of mask prompt types. The terms 'FUM', 'PAM', 'REM', and 'EPM' denote the mask types for the full image, partial image with multiple entities, dual entities' mask, and single entity mask, respectively. The checkmark symbol (✓) and the circle symbol (○) indicate whether the ablated module is used or not.

| Structure | PPP | | | COCO-PSG | | |
|---|---|---|---|---|---|---|
| | $AP^P$ | $AP^E$ | $AR^{EP}$ | $AP^R$ | $AP^E$ | $AR^{RE}$ |
| None | 24.5 | 53.4 | 69.7 | 38.9 | 40.4 | 50.9 |
| (CONV2B×2)+(CONV1) | **26.5** | **56.1** | **72.3** | **40.5** | **42.1** | **53.1** |
| (CONV2B×2)+(CONV1×2) | 26.4 | 56.1 | 72.3 | 40.5 | 42.1 | 53.1 |
| (CONV2B×2) | 25.9 | 54.7 | 71.1 | 39.6 | 41.6 | 52.5 |

(a)

| SA | FF | ADD | PPP | | | COCO-PSG | | |
|---|---|---|---|---|---|---|---|---|
| | | | $AP^P$ | $AP^E$ | $AR^{EP}$ | $AP^R$ | $AP^E$ | $AR^{RE}$ |
| ○ | ○ | ○ | 24.5 | 53.4 | 69.7 | 38.9 | 40.4 | 50.9 |
| ✓ | ○ | ○ | 25.5 | 54.8 | 70.7 | 39.1 | 40.9 | 51.7 |
| ✓ | ✓ | ○ | 25.7 | 54.3 | 70.8 | 39.7 | 41.2 | 52.3 |
| ✓ | ✓ | ✓ | **26.5** | **56.1** | **72.3** | **40.5** | **42.1** | **53.1** |

(b)

Table 4: Ablation studies. **(a)** The structure design of prompt mask encoder. The 'CONV2B' and 'CONV1' indicates convolution block operation with kernel and stride size 2 followed by layer normalization and GELU and convolution with kernel and stride size 1. The ×2 means the two sequential structures. **(b)** The structure design of the task correlation module. The terms 'SA', 'FF', and 'ADD' denote self-attention, feed-forward, and addition respectively.

levels (mAP and association recall) on the two evaluation subsets, highlighting the effectiveness of these proposed modules on all levels. With the addition of the association module, a more significant improvement in association performance can be achieved while maintaining a similar level of segmentation performance compared to the baseline. This indicates that our proposed association module does not negatively impact the segmentation performance. Furthermore, all three modules can mutually benefit each other, giving the best performance as shown in the last row.

**Mask Prompt Encoder** In Table 3(b), we demonstrate the impact of using different mask prompt types during training. Note that only full-image mask prompts are used during inference. The 1st and 2nd rows show that full-image mask prompts do not bring any impact. In the third row, solely using a mask prompt based on the partial annotation area of the image results in a significant performance decline due to the mask prompt inconsistency between the training and inference stages, because inference only uses full-image prompts. However, when both full and partial image prompts are used, as shown in the 4th row, better performance is achieved by simultaneously mitigating the incomplete annotation problem and maintaining training-inference consistency. As shown by the last row, the introduction of dual-entity or single-entity mask prompts further improves performance, by pushing the model to learn better how to split masks between two immediate adjacent levels (relation-to-entities & entity-to-parts). This is different than (but complements) using full- or partial-image mask prompts which impose large gaps between the source and target levels (*e.g.,* image-to-parts).

In Table 4(a), we examine the influence of various mask encoder architectures The 1st row represents our baseline without a prompt mask encoder. As demonstrated in the 2nd and 4th rows, we find that adding a 'CONV2' and 'CONV1' block with convolution kernel and stride sizes of 2 and 1 is sufficient for the mask encoder architecture. Incorporating more 'CONV1' blocks does not lead to a significant improvement in the final performance, as shown in the third row. This is because a single 'CONV1' layer is enough to encode mask information after the feature that has been already processed by two 'CONV2' blocks, as evident in the last two rows.

**Task Complementarity Module**. Table 4(b) presents the ablation study on the design of TCM. It is evident that the self-attention block is capable of fusing information from each level to enhance each respective task, as demonstrated in the 2nd row. Moreover, adding the original level-specific embeddings back to fused embeddings is crucial for improving the final performance. This is because the level-specific embeddings carries unique information specific to each task.

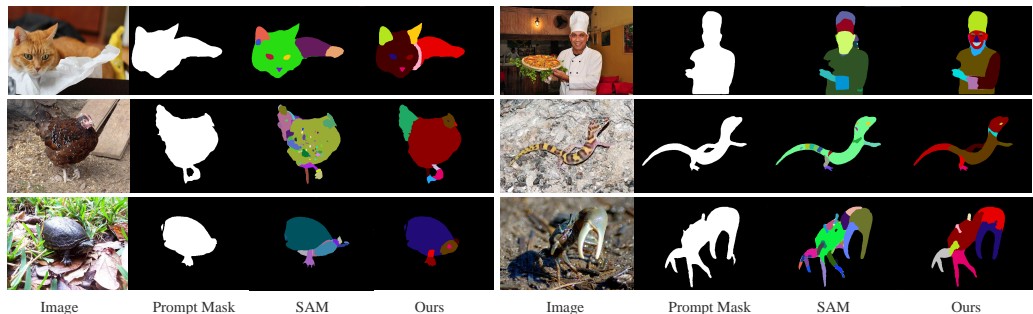

| Image | Prompt Mask | SAM | Ours | Image | Prompt Mask | SAM | Ours |

Figure 5: Qualitative comparison of SAM [10] and our method in interactive segmentation. The prompt mask indicates an entity mask that is provided to the network for further separation to the next low level. For the SAM model, we use hyper-parameters for more fine-grained segmentation, which will be introduced in our supplementary material. The images in the first row and the last two rows are from PACO [9] (in-domain) and ImageNet [56] (out-of-domain).

| Method | PPP | | | COCO-PSG | | |
|---|---|---|---|---|---|---|
| | PQ | PartPQ | PWQ | R/mR@20 | R/mR@50 | R/mR@100 |
| PanopticPartFormer++ [7] | 68.0 | 67.0 | 68.3 | - | - | - |
| PSGTR [8] | - | - | - | 28.2/15.4 | 32.1/20.3 | 35.3/21.5 |
| Ours | **72.9** | **70.8** | **72.3** | **36.7/22.9** | **39.4/25.6** | **42.3/28.2** |

(a)

| Method | Train Data | VG-150 $AR^R$ | PACO $AR^E$ | PACO $AR^P$ | ADE20K $AR^E$ | ADE20K $AR^P$ |
|---|---|---|---|---|---|---|
| EntitySeg [5] | 33K | - | 87.2 | - | 82.7 | - |
| SAM [10] | 11M | - | 86.2 | 52.3 | 82.3 | 49.7 |
| Ours | 237K | **67.4** | **87.1** | **52.9** | **83.0** | **50.4** |

(b)

Table 5: Ablation studies. **(a)** Comparison with state-of-the-art (SOTA) methods in panoptic part segmentation and panoptic scene graph tasks. **(b)** Performance comparison of segmentation generalization for images in the real world. The $AR^R$, $AR^E$, and $AR^P$ represent recall@100 at relation, entity, and part levels, respectively.

## 4.2 Comparison to state-of-the-art methods

In the following comparison, we evaluate our model against other state-of-the-art methods. To ensure a fair comparison, all methods employ Swin-Large backbone, with the exception of SAM, which uses a ViT-Huge backbone [55].

**Category-aware segmentation.** We transfer our AIMS method to other class-aware segmentation tasks, including panoptic part segmentation and panoptic scene graph on PPP and COCO-PSG datasets, respectively. To equip our method with class-aware capability, we directly use our AIMS model as pre-trained weights to fine-tune the corresponding class-aware training dataset. In Table 5(a), we observe that our AIMS model outperforms the corresponding state-of-the-art methods on both tasks. This is because our AIMS model can utilize more training data from various datasets by employing our proposed prompt mask encoder to address annotation inconsistencies among them. Moreover, our AIMS model benefits from the two-stage process of class-agnostic pretraining and class-aware fine-tuning [38], which has been demonstrated to improve the performance of the second stage class-aware fine-tuning when class-agnostic pretraining is performed with more data.

**Interactive segmentation.** In Table 5(b), we compare AIMS to another popular interactive segmentation method, SAM [10], on the PACO and ADE20K [57] datasets, where ADE20K is for zero-shot evaluation at entity and part levels. Neither SAM nor AIMS uses ADE20K for training, and ADE20K's classes are fairly dissimilar to PascalVOC's [58]. The 1st row represents our baseline method with purely class-agnostic training without interactive segmentation, which outperforms SAM at the pure entity level even though the SAM model is trained on 11 million images. This indicates that the zero-shot generalization ability is attributed to class-agnostic rather than interactive training. On the other hand, SAM's interactive segmentation can divide masks into more fine-grained segments, which our AIMS is also capable of. On both evaluation datasets, AIMS remarkably outperforms SAM, using merely a 273K training set that is far smaller than SAM's 11M training set. Our proposed MPE and TCM allow our model to leverage more desirable training signals and multi-level task information in a data-efficient manner. Furthermore, our AIMS model can make relation-level predictions, which are higher than part and entity levels and absent in the SAM model.

Figure 5 provides a visual comparison between the SAM [10] model and our model in a real-world scenario. Like AIMS, the SAM model also supports image segmentation at three levels (object,

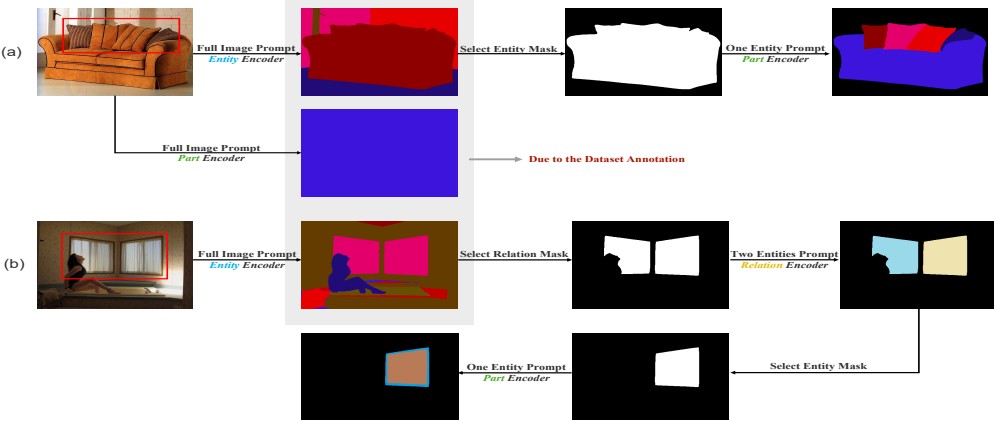

Figure 6: The illustration of the flexibility of AIMS tasks to tackle the subjective annotation issues in existing datasets.

part, and sub-part level). For an equitable comparison, we use the same entity mask prompt for further delineation. As shown in Figure 5, both models capably segment parts of the entity with distinct approaches. The SAM model leans towards identifying parts based on color and texture clues, whereas our AIMS model relies more on semantic understanding.

**The flexibility of our AIMS task.** Figure 7 illustrates how our proposed AIMS task provides the flexibility for segmenting anything. This is to address the subjective annotation issues in existing datasets. For instance, in the image (a), the throw pillows and the sofa are predicted as a single entity, which aligns with the ground truth annotation. Nonetheless, in certain scenarios, a user might wish to edit the throw pillows independently. With our AIMS method, utilizing a full image prompt for part-level prediction does not yield these separated masks. However, by selecting this entity mask as an entity prompt for part-level prediction, we can successfully differentiate the three throw pillows.

Additionally, in image (b), a user may want to segment the windows into several components. However, original ground truth annotations typically consider the two windows as a single entity. To meet this requirement, we initially utilize a full image prompt and an entity encoder to identify the mask of the whole windows. Following this, we apply this two-entity mask prompt and relation Decoder to split them into two independent window masks. Ultimately, we can select one window for further segmentation into two parts: the window edge and curtains.

# 5 Conclusion

This paper presents AIMS, a unified segmentation model that parses images at three levels: part, entity, and relation. To address the absence of a dataset with annotations for all three levels, we construct a unified dataset by aggregating multiple existing datasets, each providing annotations for one or two levels. Our base method uses three separate decoders for different-level predictions. To handle the annotation inconsistency issue, we propose Mask Prompt Encoder, which offers a more accurate task signal to the model, and Task Complementarity Module to enhance each level's prediction. Extensive experiments demonstrate the effectiveness of our method in both closed-world and zero-shot settings. Our work can serve as a springboard for further research on this new segmentation problem.

# 6 Acknowledgement

This research is based upon work supported by the Office of the Director of National Intelligence (ODNI), Intelligence Advanced Research Projects Activity (IARPA), via IARPA R&D Contract No. 2022-21102100001. The views and conclusions contained herein are those of the authors and should not be interpreted as necessarily representing the official policies or endorsements, either expressed or implied, of the ODNI, IARPA, or the U.S. Government. The U.S. Government is authorized to reproduce and distribute reprints for Governmental purposes notwithstanding any copyright annotation thereon.

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

# Appendix

In this supplementary file, we provide more experimental details and empirical results to further demonstrate the benefits of our proposed All-Inclusive Multi-Level segmentation task (AIMS).

- Experimental details on dataset split, sampling strategy and SAM's [10] configuration.
- More ablation studies on sample strategy and dataset usage.
- More visualization results of our AIMS model with various inference modes.
- User study of AIMS and the concurrent work SAM [10]

The code and models to reproduce our experiments will be released.

## A  Experimental Details

**Dataset Split**    As outlined in the main paper, all models are trained using five datasets: COCO [53], EntitySeg [5], PascalVOC Part (PPP)[6], PACO[9], and COCO-PSG [8]. Given that the original training and validation splits of these datasets are tailored for single tasks, we collate the images and reorganize them to suit our AIMS task. Initially, we select 1069 and 1000 validation images from PPP [6] (which covers the part and entity levels) and COCO-PSG [8] (which covers the entity and relation levels) respectively. Following this, we eliminate any duplicate images in the unified training set that are present in the validation images, resulting in a refined training set comprised of approximately 236.7K unique images.

**Sampling Strategy**    In Table 7, we present the eight different sample types, labeled as Sample ID 1 through 8, for each iteration, assuming a batch size of 8. Additionally, images for each of these sample types are uniformly selected from the datasets mentioned in the 'Dataset' column.

**SAM's configuration**    To obtain more fine-grained part-level predictions, we follow the hyper-parameters found on SAM's GitHub page: *points_per_side* (2), *pred_iou_thresh* (0.86), *stability_score_thresh* (0.92), *crop_n_layers* (1), *crop_n_points_downscale_factor* (2), *min_mask_region_area* (100).

## B  Ablation Studies

**Sample Strategy**    Table 7 presents an ablation study on various sample strategies by examining the impact of each sample type during each iteration. The first row serves as our baseline, adhering to our base framework with a split decoder for varying-level predictions, and it excludes the usage of a prompt mask encoder. The remaining rows illustrate the influence on performance when task prompts are employed during each training iteration.

For instance, the second row reveals that the exclusive usage of a full image task prompt brings a marginal performance improvement over the baseline. Since full-image mask prompts are always the same, injecting them into the original image features is identical to not using any mask prompts. Conversely, the third row shows that employing solely partial-image mask prompts can considerably deteriorate performance, which is inconsistent with the inference process that involves full-image mask prompts.

In the fourth row, we observe that using both full and partial mask prompts can further enhance performance by providing more accurate signals to the network. Additionally, the introduction of mask prompts at the entity-to-part and relation-to-entity levels consistently yields performance improvements, as depicted in subsequent rows.

**Dataset Usage**    Table 8 displays an ablation study focusing on the utilization of various datasets. The first three rows present the model performance when trained on a single dataset. This results in degraded performance on cross-dataset evaluation due to inconsistencies in annotations across different datasets. As shown in the fourth row, combining the three datasets for training enhances the overall validation performance. This improvement is due to the consistency maintained between the training and validation splits. Incorporating PACO and Entityseg datasets at part and entity levels further improve the performance for the respective levels.

| Sample ID | Mask Prompt | Decoder | Dataset |
|---|---|---|---|
| 1 | Full Image | Entity | COCO, EntitySeg, PPP, COCO-PSG |
| 2 | | Part | PPP |
| 3 | | Relation | COCO-PSG |
| 4 | Partial Image | Entity | COCO, EntitySeg, PPP, COCO-PSG, PACO |
| 5 | | Part | PPP, PACO |
| 6 | | Relation | COCO-PSG |
| 7 | One Entity | Part | PPP, PACO |
| 8 | Two Entities | Relation | COCO-PSG |

Table 6: The illustration of eight sample types on each training iteration.

| Sample ID | | | | | | | | PPP (Inference) | | | COCO-PSG (Inference) | | |
|---|---|---|---|---|---|---|---|---|---|---|---|---|---|
| 1 | 2 | 3 | 4 | 5 | 6 | 7 | 8 | $AP^P$ | $AP^E$ | $AR^{EP}$ | $AP^R$ | $AP^E$ | $AR^{RE}$ |
| ○ | ○ | ○ | ○ | ○ | ○ | ○ | ○ | 24.5 | 53.4 | 69.7 | 38.9 | 40.4 | 50.9 |
| ✓ | ✓ | ✓ | ○ | ○ | ○ | ○ | ○ | 25.0 | 53.5 | 69.8 | 39.2 | 40.8 | 51.3 |
| ○ | ○ | ○ | ✓ | ✓ | ✓ | ○ | ○ | 22.5 | 51.4 | 67.3 | 37.1 | 37.6 | 48.9 |
| ✓ | ✓ | ✓ | ✓ | ✓ | ✓ | ○ | ○ | 25.7 | 54.4 | 70.9 | 39.6 | 41.2 | 51.8 |
| ✓ | ✓ | ✓ | ✓ | ✓ | ✓ | ✓ | ○ | 26.4 | 55.9 | 72.1 | 39.6 | 41.3 | 51.8 |
| ✓ | ✓ | ✓ | ✓ | ✓ | ✓ | ○ | ✓ | 25.7 | 54.3 | 71.0 | 40.4 | 42.0 | 53.0 |
| ✓ | ✓ | ✓ | ✓ | ✓ | ✓ | ✓ | ✓ | **26.5** | **56.1** | **72.3** | **40.5** | **42.1** | **53.1** |

Table 7: The ablation studies on eight distinct sample types used in each training iteration. The ✓ and ○ symbols are employed to indicate whether a specific sample type is utilized. For a fair comparison, we adjust the learning rate linearly in relation to the batch size. The default learning rate is 1e-8 for a batch size of 8.

| Dataset | | | | | PPP (Inference) | | | COCO-PSG (Inference) | | |
|---|---|---|---|---|---|---|---|---|---|---|
| COCO | COCO-PSG | PPP | PACO | EntitySeg | $AP^P$ | $AP^E$ | $AR^{EP}$ | $AP^R$ | $AP^E$ | $AR^{RE}$ |
| ✓ | ○ | ○ | ○ | ○ | - | 44.4 | - | - | 41.2 | - |
| ○ | ✓ | ○ | ○ | ○ | - | 44.5 | - | 39.6 | 41.7 | 51.8 |
| ○ | ○ | ✓ | ○ | ○ | 24.3 | 48.6 | 65.8 | - | 25.7 | - |
| ✓ | ✓ | ✓ | ○ | ○ | 25.7 | 54.4 | 70.9 | 39.6 | 41.2 | 51.8 |
| ✓ | ✓ | ✓ | ✓ | ○ | 26.4 | 55.9 | 72.1 | 39.6 | 41.2 | 51.8 |
| ✓ | ✓ | ✓ | ○ | ✓ | 25.7 | 54.3 | 71.0 | 40.4 | 42.0 | 53.0 |
| ✓ | ✓ | ✓ | ✓ | ✓ | **26.5** | **56.1** | **72.3** | **40.5** | **42.1** | **53.1** |

Table 8: The ablation study of performance influence with dataset usage. Similar to Table 7, we adjust the learning rate linearly considering the lack of some task prompts due to the non-provided dataset.

# C  Visualization

**The flexibility of our AIMS task.**    Figure 7 illustrates how our proposed AIMS task provides the flexibility for segmenting anything. This is to address the subjective annotation issues in existing datasets. For instance, in the image (a), the throw pillows and the sofa are predicted as a single entity, which aligns with the ground truth annotation. Nonetheless, in certain scenarios, a user might wish to edit the throw pillows independently. With our AIMS method, utilizing a full image prompt for part-level prediction does not yield these separated masks. However, by selecting this entity mask as an entity prompt for part-level prediction, we can successfully differentiate the three throw pillows.

Additionally, in image (b), a user may want to segment the windows into several components. However, original ground truth annotations typically consider the two windows as a single entity. To meet this requirement, we initially utilize a full image prompt and an entity encoder to identify the mask of the whole windows. Following this, we apply this two-entity mask prompt and relation Decoder to split them into two independent window masks. Ultimately, we can select one window for further segmentation into two parts: the window edge and curtains.

**Prediction consistency.**    We investigate the prediction consistency across two inference modes: full and partial image prompts. Figure 8 displays an example where our AIMS model fails to segment anything at the part level with a full image prompt, but it is able to break down an entity-level mask prompt into a more detailed level, similar to the sofa and throw pillows in the image. Given that the pillows have never been labeled in our utilized dataset, we surmise that our AIMS model might yield inconsistent prediction results for unseen classes when different prompts are used. Using entity-level mask prompts could help our model to drill down to the next level.

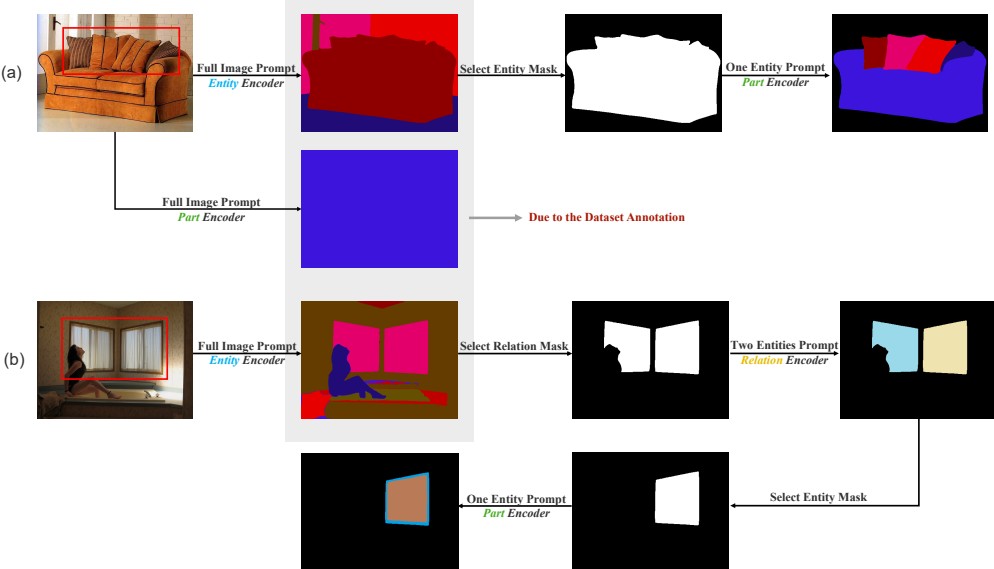

Figure 7: The illustration of the flexibility of AIMS task to tackle the subjective annotation issues in existing datasets.

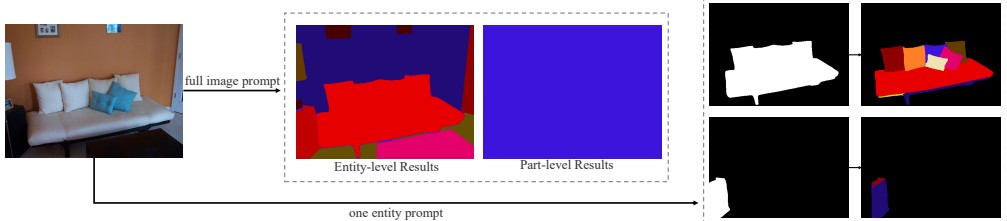

Figure 8: The example of prediction inconsistency between two prompt types on unseen classes.

However, the scenario changes when we turn our attention to the known classes. Figure 9 illustrates that three cars yield similar part-level prediction results, regardless of whether a full image or a single entity prompt is used. This indicates that our AIMS model can maintain prediction consistency for known classes.

**The three-level predictions**   Figure 10 displays the three-level prediction outcomes of our AIMS model. Despite the potential subjectivity in defining the three levels, our AIMS model shows strong promises in fulfilling various user needs and intentions in image editing. For instance, the red region in the image is not detected at the entity-level prediction results, as it is merged into the tray. However, by injecting an entity-level mask prompt for the tray region into the part encoder, we see that the separate red region can be obtained, as shown in the second column of the 'part-level results'. Additionally, the relation-level masks of any two entities that share some semantic relationship can be predicted, and all relation-level prediction results can be represented in a scene graph.

**More visualization results.**   In the Fig. 13, we show more visualization results at entity and part level in the wild (Open-Image [59] dataset), manifesting the generalization ability of our proposed AIMS model at the entity and part level. Considering the entity level is the principal part of image editing, we also show some cases on Laion400M [60] to show the effectiveness of our AIMS model.

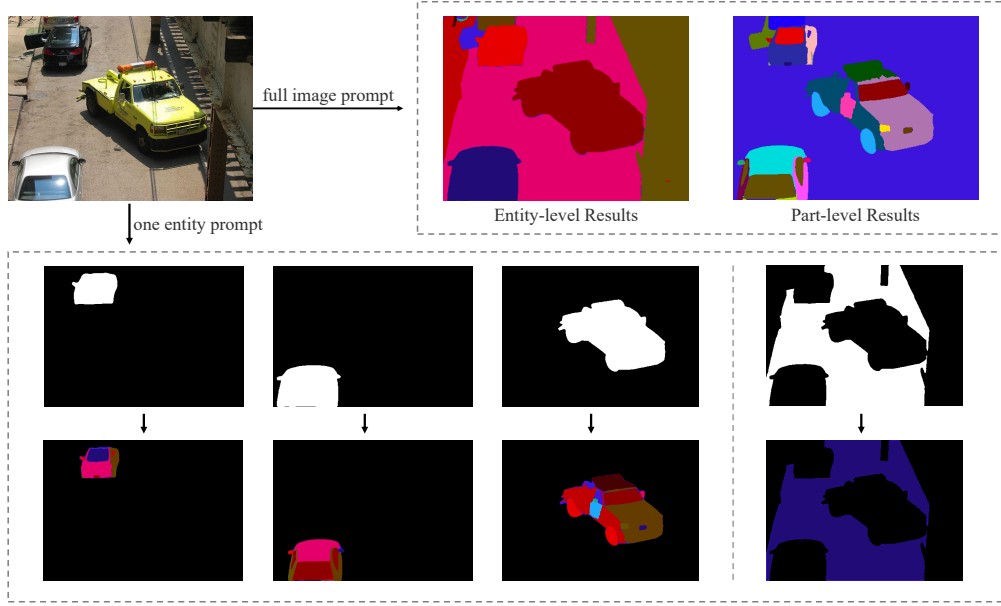

Figure 9: The example of prediction consistency between two prompt types on seen classes.

## D    User Study

We conducted a user study in which there were 480 individuals who were identified as Adobe Photoshop users who regularly used the software for image manipulation/editing. In this user study, we randomly selected 40 images in the wild and provided the users with a visualization of the three-level prediction results of our AIMS model, as shown in Figure 10. For each image, we asked each user about his/her degree of satisfaction with treating semantically meaningful and -coherent segments for three levels, with respect to their relevance to and suitability for image manipulation/editing applications. The satisfactory scores are aggregated from all users for the individual images. We find that the average score of each image is large than 7.8 on the condition that the maximum score is 10. Most of the selected images' scores are larger than 6.0, This confirms that the users are highly satisfied with our model's prediction results in the context of image manipulation/editing. To better present the user study's findings, we summarize the data of Table 9. The image IDs with the minimum, median, and maximum user scores are 20, 19, and 35. We show these images and their corresponding entity annotations in Fig. 12.

In Fig. 12, we provide statistical information about our user study. The sub-figure (a) is a summary of Table 9, indicating that over 90% of users rated our prediction results higher than 6.0. Sub-figure (b) contrasts user preferences between our results and those from the SAM model. Despite being trained with fewer data, our model garners an appreciation comparable to that of SAM, as can be observed in the sub-figure (b).

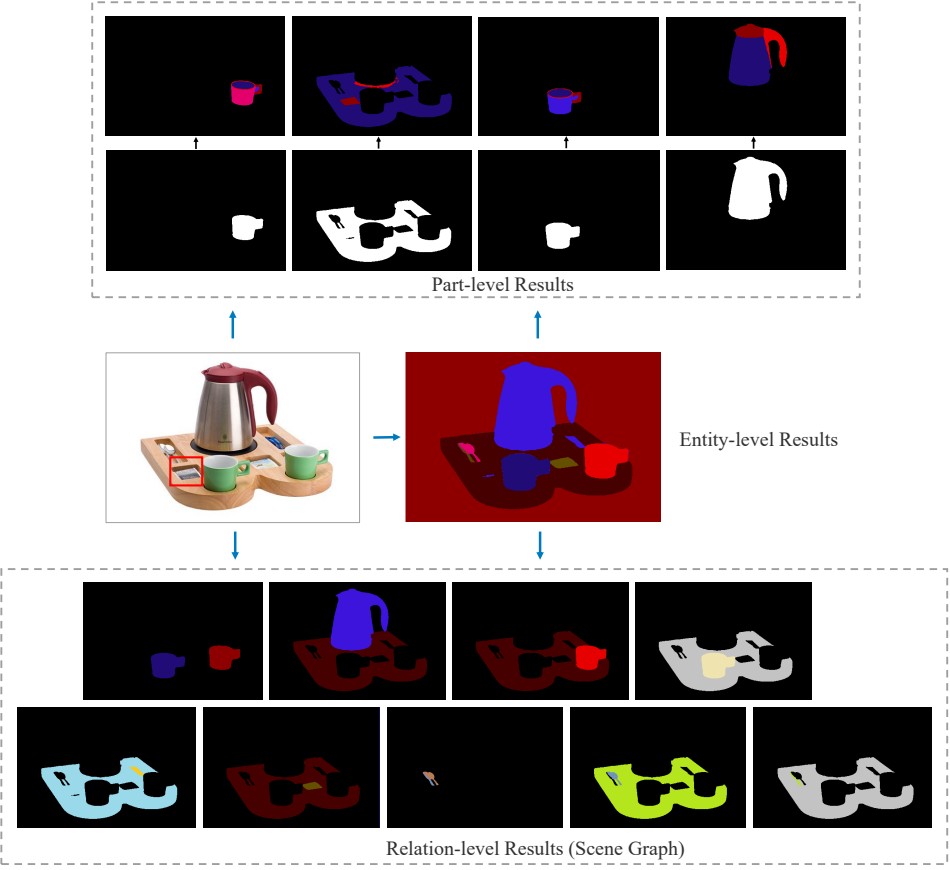

Figure 10: The illustration of how the AIMS model can be used to obtain multi-intention segmentation results, including a scene graph for physical-touch relations.

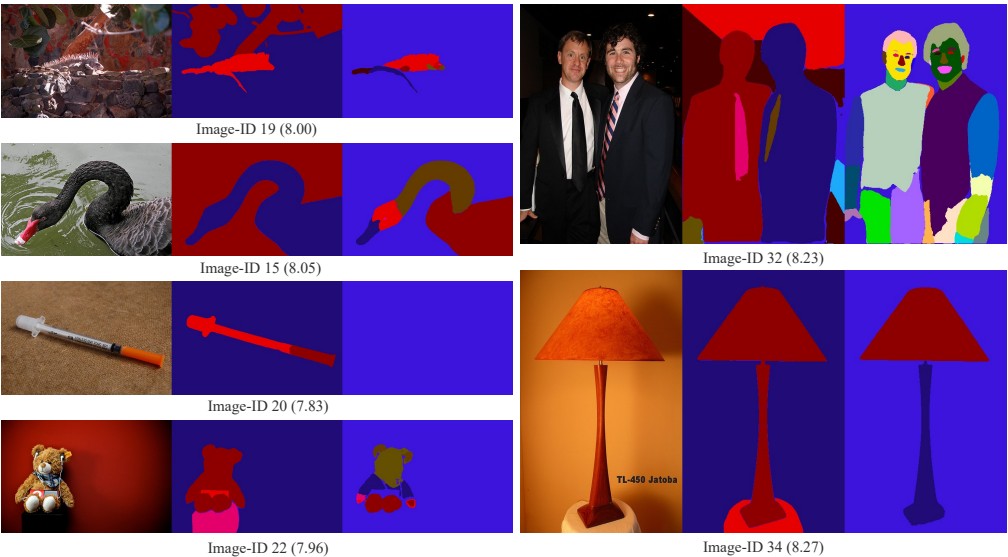

Figure 11: The selected images used in our user study. For brevity, we only show the entity- and part-level predictions the user most care.

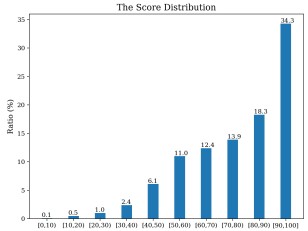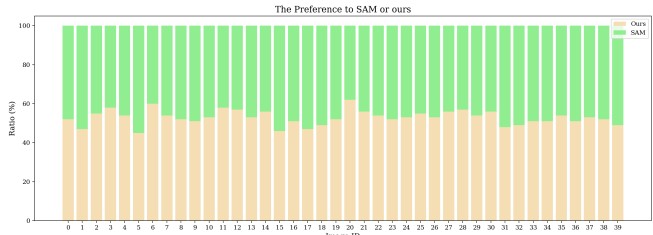

Figure 12: The statistical visualizations of the data from our user and survey studies. **(a)** The histogram here represents the distribution of the users based on their given scores in the three-level predictions. **(b)** Each horizontal bar here indicates the proportions of votes given by the survey participants to SAM [10] (green) and *ours* (orange) on each of the 40 images.

| Image ID | $P_1$ | $P_2$ | $P_3$ | $P_4$ | $P_5$ | $P_6$ | $P_7$ | $P_8$ | $P_9$ | $P_{10}$ | AVG |
|---|---|---|---|---|---|---|---|---|---|---|---|
| 1 | 0.00 | 0.00 | 1.88 | 3.13 | 8.13 | 11.25 | 11.25 | 16.88 | 9.38 | 38.13 | 8.06 |
| 2 | 0.63 | 0.00 | 3.13 | 3.75 | 5.63 | 15.00 | 13.75 | 13.13 | 10.00 | 35.00 | 7.84 |
| 3 | 1.25 | 0.63 | 1.88 | 1.88 | 10.63 | 10.63 | 15.00 | 12.50 | 8.13 | 37.50 | 7.86 |
| 4 | 0.63 | 0.63 | 0.63 | 3.13 | 7.50 | 14.38 | 13.13 | 14.38 | 8.75 | 36.88 | 7.95 |
| 5 | 0.00 | 0.63 | 2.50 | 1.25 | 7.50 | 11.88 | 13.13 | 18.13 | 7.50 | 37.50 | 8.02 |
| 6 | 0.63 | 0.63 | 1.25 | 1.25 | 6.88 | 15.00 | 14.38 | 20.00 | 7.50 | 32.50 | 7.88 |
| 7 | 0.00 | 0.00 | 0.63 | 3.13 | 6.25 | 13.75 | 12.50 | 20.63 | 10.63 | 32.50 | 8.01 |
| 8 | 0.00 | 0.00 | 1.25 | 1.88 | 8.13 | 11.88 | 15.63 | 13.75 | 9.38 | 38.13 | 8.08 |
| 9 | 0.00 | 0.63 | 1.25 | 3.75 | 8.13 | 10.63 | 15.63 | 18.13 | 8.13 | 33.75 | 7.90 |
| 10 | 0.00 | 0.63 | 3.13 | 2.50 | 5.00 | 16.88 | 15.00 | 14.38 | 6.88 | 35.63 | 7.85 |
| 11 | 0.00 | 0.00 | 0.00 | 1.25 | 5.63 | 17.50 | 10.00 | 19.38 | 13.75 | 32.50 | 8.12 |
| 12 | 0.63 | 0.63 | 0.63 | 2.50 | 5.00 | 10.63 | 16.88 | 22.50 | 11.87 | 31.88 | 8.00 |
| 13 | 0.00 | 0.00 | 0.00 | 2.50 | 5.00 | 14.38 | 15.00 | 15.63 | 12.50 | 35.00 | 8.14 |
| 14 | 0.00 | 0.63 | 1.25 | 3.75 | 6.25 | 13.75 | 13.75 | 19.38 | 12.50 | 28.75 | 7.85 |
| 15 | 0.00 | 0.00 | 0.63 | 1.88 | 8.13 | 13.75 | 11.25 | 20.63 | 9.38 | 34.38 | 8.05 |
| 16 | 0.63 | 0.63 | 1.25 | 2.50 | 7.50 | 11.88 | 13.75 | 18.13 | 11.25 | 32.50 | 7.91 |
| 17 | 0.00 | 0.00 | 0.63 | 1.88 | 4.38 | 13.75 | 18.13 | 16.25 | 11.88 | 33.13 | 8.09 |
| 18 | 0.00 | 0.00 | 0.00 | 3.13 | 5.63 | 16.88 | 15.00 | 13.75 | 11.25 | 34.38 | 8.02 |
| 19 | 0.00 | 0.63 | 0.63 | 1.88 | 6.25 | 15.00 | 16.88 | 14.38 | 8.75 | 35.63 | 8.00 |
| 20 | 0.00 | 0.00 | 0.63 | 3.13 | 6.25 | 10.63 | 12.50 | 20.63 | 10.63 | 32.50 | 7.83 |
| 21 | 0.00 | 1.88 | 0.63 | 0.63 | 5.00 | 15.63 | 15.63 | 22.50 | 8.13 | 33.75 | 8.05 |
| 22 | 0.00 | 0.63 | 1.25 | 3.13 | 8.75 | 10.63 | 13.75 | 16.25 | 11.88 | 33.75 | 7.96 |
| 23 | 0.00 | 0.63 | 0.63 | 1.88 | 5.63 | 13.75 | 13.75 | 18.75 | 9.38 | 35.63 | 8.08 |
| 24 | 0.00 | 0.63 | 0.63 | 1.88 | 4.38 | 11.88 | 15.63 | 20.00 | 11.88 | 33.13 | 8.11 |
| 25 | 0.00 | 1.88 | 0.63 | 1.88 | 7.50 | 11.88 | 11.88 | 19.38 | 11.25 | 33.75 | 8.00 |
| 26 | 0.00 | 1.25 | 0.63 | 2.50 | 5.00 | 11.25 | 16.88 | 18.75 | 9.38 | 34.38 | 8.03 |
| 27 | 0.00 | 0.00 | 1.25 | 1.25 | 7.50 | 12.50 | 13.13 | 20.63 | 9.38 | 34.38 | 8.06 |
| 28 | 0.00 | 0.00 | 0.63 | 4.38 | 3.75 | 12.50 | 13.13 | 20.00 | 13.13 | 32.50 | 8.08 |
| 29 | 0.00 | 0.00 | 0.63 | 3.13 | 5.00 | 8.13 | 16.88 | 18.75 | 13.75 | 33.75 | 8.18 |
| 30 | 0.00 | 1.25 | 1.25 | 5.00 | 5.63 | 11.25 | 11.25 | 18.13 | 11.88 | 34.38 | 7.96 |
| 31 | 0.00 | 0.63 | 1.88 | 0.63 | 5.00 | 12.50 | 12.50 | 20.00 | 12.50 | 34.38 | 8.13 |
| 32 | 0.00 | 0.63 | 1.88 | 0.00 | 4.38 | 11.25 | 15.00 | 18.75 | 9.38 | 38.75 | 8.23 |
| 33 | 0.00 | 0.63 | 0.63 | 2.50 | 5.00 | 13.13 | 9.38 | 23.75 | 9.38 | 35.63 | 8.13 |
| 34 | 0.00 | 1.25 | 0.63 | 2.50 | 5.00 | 11.88 | 6.88 | 18.75 | 15.63 | 37.50 | 8.25 |
| 35 | 0.00 | 0.63 | 0.63 | 1.25 | 4.38 | 10.00 | 12.50 | 21.88 | 13.13 | 35.63 | 8.27 |
| 36 | 0.00 | 0.63 | 0.63 | 3.13 | 5.00 | 10.63 | 13.13 | 18.13 | 14.38 | 34.38 | 8.14 |
| 37 | 0.00 | 0.00 | 0.63 | 2.50 | 3.75 | 10.00 | 18.13 | 18.75 | 16.25 | 30.00 | 8.14 |
| 38 | 0.63 | 0.00 | 0.63 | 2.50 | 10.00 | 10.63 | 13.13 | 23.13 | 10.00 | 29.38 | 7.84 |
| 39 | 0.00 | 1.25 | 1.88 | 1.88 | 4.38 | 10.63 | 18.13 | 17.50 | 13.75 | 30.63 | 8.04 |
| 40 | 0.00 | 0.00 | 0.63 | 2.50 | 6.88 | 11.88 | 11.25 | 16.88 | 13.75 | 36.25 | 8.08 |

Table 9: Data from the user study on the three-level prediction results of our AIMS model. $P_x$ indicates the percentage of users who give "x" as the score to represent their degrees of satisfaction. "x" ranges from 1 to 10, and 10 represents the highest degree of satisfaction. "AVG" indicates the the score averaged across all users for each image.

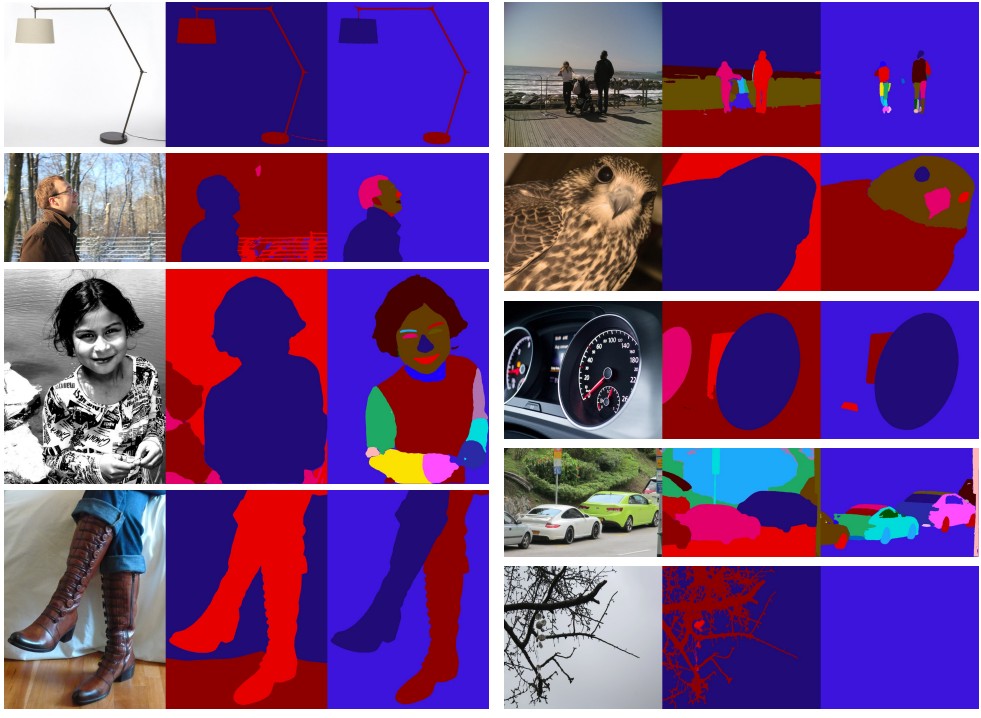

Figure 13: More visualization results on Open-Image Dataset [59] in the 'wild'.

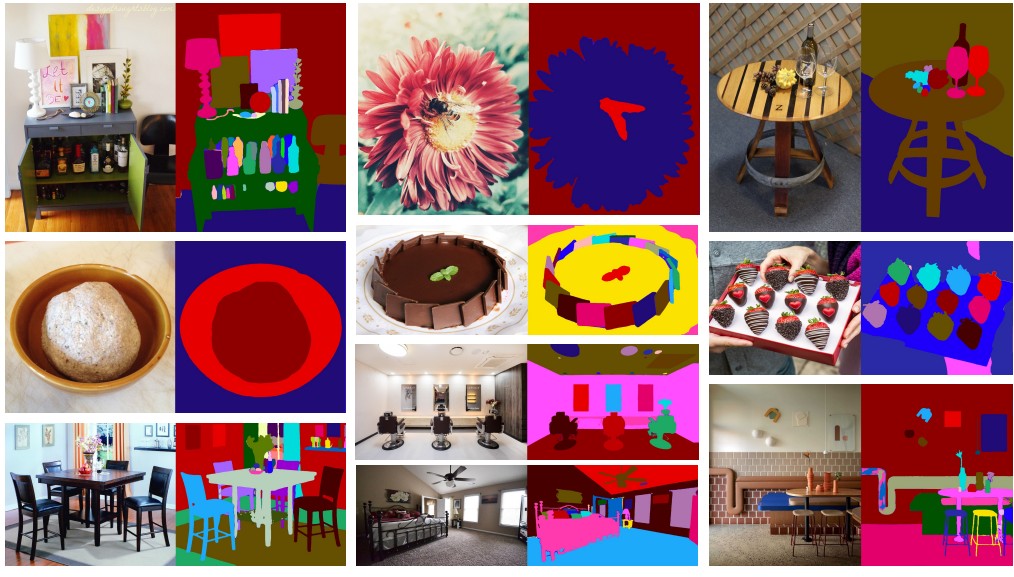

Figure 14: More visualization results on Laion400M [60] in the 'wild'.

