# OpenReview forum: "AIMS: All-Inclusive Multi-Level Segmentation for Anything"
_NeurIPS.cc/2023/Conference — NeurIPS 2023 spotlight_

### Official Review · Reviewer_S47w · 2023-06-16

**Soundness:** 3 good
**Presentation:** 3 good
**Contribution:** 3 good
**Rating:** 8
**Confidence:** 5

**Summary:**

This paper proposes a new task that segments visual regions into three levels: part, entity, and their relation. The authors build a unified AIMS model through multi-dataset multi-task training to address the two major challenges of annotation inconsistency and task correlation. Extensive experiments show the effectiveness of proposed approach.

Overall, this paper is good to me in case of new concepts and unified modeling.

**Strengths:**


-	The proposed AIMS task is interesting and novel to me, which unify several recent tasks, including entity segmentation, panoptic-part segmentation, and panoptic scene graph generation. It is a trend to unify multi-level and multi-dataset segmentation using a transformer architecture.

-	The proposed module including Task Correlation Module and R-E Complementarity Module are proven effectively in experiment part.

-	The design of mask prompt is interesting for multi-dataset segmentation and part-whole segmentation.  The proposed method of multi-dataset multi-task training is novel. Especially the method proposes the mask prompt encoder to solve the incomplete annotations on three levels among datasets. This encoder can receive eight mask prompts and specify the corresponding region slated for segmentation. This way, the proposed method can relieve the annotation ambiguity among datasets.

-	The ablation studies are good. The performance looks good compared with recent PanopticPartFormer++ and PSGTR. It also shows better generation than SAM.



**Weaknesses:**



- Presentation issues:
   Several citations are missing. For example, L-118, “a masked cross-attention, self-attention, and feedforward network”，To be my knowledge, masked cross-attention is proposed by Mask2Former, it is better add a citation to the origin work. Also, is the pixel decoder the default decoder in Mask2Former? Moreover, several details are missing. It is hard to know the benefits

- Several papers are missing:
	-Part-whole modeling paper:

		[1], Hierarchical Human Parsing with Typed Part-Relation Reasoning,CVPR-2020

		[2], Differentiable Multi-Granularity Human Representation Learning for Instance-Aware Human Semantic Parsing,CVPR-2021
        Both works aims at solving part-whole segmentation, which is close to this work.

	-Multi-task query interaction paper:

		[1], Fashionformer: A simple, Effective and Unified Baseline for Human Fashion Segmentation and Recognition, ECCV-2022

		[2], PolyphonicFormer: Unified Query Learning for Depth-aware Video Panoptic Segmentation, ECCV-2022.


	Both works adopt multiple task queries for joint reasoning and attention, which is similar to the proposed method in this work.

        Moreover, some related works to unsupervised grouping. For example,
        [1], Unsupervised Hierarchical Semantic Segmentation with Multiview Cosegmentation and Clustering Transformers. CVPR2022.

        [2], LASSIE:  Learning Articulated Shapes from Sparse Image Ensemble via 3D Part Discovery. NeurlPS2022.

    These works should be discussed in the related work part. Following the last item, the writing of this work can be improved in the next drat.

- The teaser could be a little fancier since it introduces a new concept. For example, it would be better to show the difference with previous segmentation settings.

- Following the last item, an overview of different property settings including panoptic segmentation, entity segmentation, multi-dataset training, prompting can be compared in one table.

- Is this possible to obtain the real results of PACO (MAP_{f1})? If so, how to extend this transformer architecture to achieve that.

- Lacking a more comprehensive ablation study on the proposed modules (Table 3a). The authors only evaluated a few combinations. For example, what is the performance if MCE and TCM are present but AM is not?



**Questions:**

Also, I still have some questions that require the author’s clarification.
1, Could the author report the performance influence of using different architectural choices for Mask Prompt Encoder?

2, Could the author split the part-level datasets into seen and unseen categories and test the model’s generalization ability to unseen categories?

3. There are SAM models with different model sizes. Which SAM model is used for comparison?

Overall, I appreciate and like the paper’s ideas and solution for all-inclusive multi-level segmentation. The proposed method can solve the annotation inconsistency among various datasets. Compared to using many annotators to label large-scale images at three levels, the AIMS method is more efficient in utilizing multiple datasets with different-level annotations. The paper does suffer from some weaknesses, but I think its benefits outweigh its weaknesses. I hope to see those issues addressed by the authors in the rebuttal.

---

> ### Author Rebuttal · Authors · 2023-08-09
>
> We thank the reviewer for the positive feedback and insightful comments.
>
> ### Q1: Clarity issue on the decoder.
>
> We adhere to the transformer decoder design in the Mask2Former method, encompassing a set of learnable queries and nine transformer blocks. Each block is composed of a cross-attention layer, a self-attention layer, and a feed-forward network. In our AIMS decoder, we begin with three distinct transformer decoders designated for different levels of segmentation results, forming our baseline approach.
>
> Building upon this foundation, we introduce the task complementarity module to facilitate information fusion between the various levels, such as entity-part and relation-entity decoders. This innovative design was inspired by our observation that separated transformer decoders consistently outperform a shared transformer decoders, a finding that is demonstrated in our table. Essentially, this implies that different level decoders benefit from having unique features distinct from each other.
>
>
> ---
> ### Q2: Missing some related works.
>
> We will include the following related works and discuss them: Papers [1] and [2] propose a hierarchical segmentation structure for part- and entity-level segmentation, but they are exclusively targeted at human parsing, not general objects. Although [3] and [4] utilize separate queries to enhance the results of different tasks, they cannot associate mask predictions of multiple levels and prompt the outcomes. Furthermore, some unsupervised methods, such as those proposed in [5,6], group the pixels using an unsupervised clustering approach. Despite their innovative approach, they still have a significant performance gap compared to supervised methods.
>
> [1] Hierarchical Human Parsing with Typed Part-Relation Reasoning, CVPR 2020.
>
> [2] Differentiable Multi-Granularity Human Representation Learning for Instance-Aware Human Semantic Parsing, CVPR 2021.
>
> [3] Fashionformer: A Simple, Effective and Unified Baseline for Human Fashion Segmentation and Recognition, ECCV 2022.
>
> [4] PolyphonicFormer: Unified Query Learning for Depth-aware Video Panoptic Segmentation, ECCV 2022.
>
> [5] Unsupervised Hierarchical Semantic Segmentation with Multiview Cosegmentation and Clustering Transformers. CVPR 2022.
>
> [6] LASSIE: Learning Articulated Shapes from Sparse Image Ensemble via 3D Part Discovery. NeurlPS 2022.
>
>
> ---
> ### Q3: Changing the teaser figure for better illustration.
>
> Please check figure 1 in our submitted pdf to manifest our multi-level targets against the single-level segmentation tasks.
>
>
> ---
> ### Q4: Adding the comparison to previous segmentation settings.
>
> The table below summarizes the comparison of various settings, illustrating that our AIMS model encompasses all three levels.
>
> | Task | relation-level | entity-level | part-level |
> | :----: | :----: | :----: | :----: |
> | Part Segmentation | &cross; | &cross; | &check; |
> | Panoptic/Instance/Entity Segmentation | &cross; | &check; | &cross; |
> | Scene Graph | &check; | &cross; | &cross; |
> | Segment Anthing | &cross; | &check; | &check; |
> | AIMS (Ours) | &check; | &check; | &check; |
>
>
> ---
> ### Q5: More results on PACO.
> We fine-tune our pretrained AIMS model directly on the class-aware PACO dataset, achieving 45.6 $AP^{obj}$ and 18.9 $AP^{opart}$. This compares favorably with the 43.4 $AP^{obj}$ and 17.7 $AP^{opart}$ obtained using the cascaded ViT-L FPN as reported in [7]. These results suggest that our AIMS model may serve as a superior pretraining model for fine-tuning in downstream tasks.
>
> [7] PACO: Parts and Attributes of Common Objects. arXiv 2023.
>
>
> ---
> ### Q6: Lacking an ablation study in Table 3(a).
>
> The table below presents an ablation study of the association module. It can be observed that the association module primarily enhances the association performance, having only a positive impact on the segmentation results. This finding aligns with the observations made in Table 3(a), where only slight variations in performance were noted. The values reported in PPP dataset are $AP^P$, $AP^E$ and $AR^{ER}$. The values reported in COCO-PSG dataset are $AP^R$, $AP^E$ and $AR^{RE}$.
>
> | Design | PPP | COCO-PSG |
> | ----- | ---- | ---- |
> | Baseline | 24.5/53.4/69.7 | 38.9/40.4/50.9 |
> | Basline with TCM and MPE | 26.4/55.9/71.4 | 40.5/42.0/52.6 |
> | Basline with TCM, MPE and AM | 26.5/56.1/72.3 | 40.5/42.1/53.1 |
>
>
> ---
> ### Q7: The ablation study on mask prompt encoder structure.
>
> We ablate the structure design of the mask prompt encoder in Table 4(a) of our paper.
>
>
> ---
> ### Q8: The experiments on part segmentation with seen/unseen data split.
>
> We reorganized the PACO and PPP datasets, dividing them into 68 seen and 10 unseen object categories, and then trained only on the part with seen objects. In the following table, we compare the part segmentation results between our method and SAM on the validation dataset, including both seen and unseen categories. It becomes evident from the comparison that our model achieves performance comparable to SAM's, but with the advantage of utilizing much less training data.
>
> | Method | seen (AR@100) | unseen (AR@100) |
> | :-----: | :----: | :----: |
> | SAM | 46.6 | 46.2 |
> | Ours | 47.0 | 46.4 |
>
>
> ---
> ### Q9: Clarity of using the SAM model.
>
> For the SAM model, we use the one with a VIT-Huge backbone, which is much larger than the Swin-Large backbone our model use.
>
> For the computation cost, conducting full image inference under (800, 1333) image size in three levels requires 854.0 GFlops and takes approximately 0.142 seconds per image on A100 40G. This computation cost is comparable to the baseline method, which consumes about 801.0 GFlops and has an inference time of 0.138 seconds per image. For the SAM model we compare, it would cost 0.253 seconds per image within the same setting.

---

> ### Author Response · Authors · 2023-08-14
> **Please let us know whether you have additional questions after reading our response**
>
> We appreciate your reviews and comments. We hope our responses address your concerns. Please let us know if you have further questions after reading our rebuttal.
>
> We hope to address all the potential issues during the discussion period.
>
> Thank you.

---

> > ### Comment · Reviewer_S47w · 2023-08-15
> > **The authors solve my issues.**
> >
> > After I see other comments and the rebuttal, the authors provided solid results. Thus, I raise my core to 8.

---

### Official Review · Reviewer_Tt8E · 2023-07-01

**Soundness:** 2 fair
**Presentation:** 2 fair
**Contribution:** 1 poor
**Rating:** 6
**Confidence:** 4

**Summary:**

This paper introduces a new task and a model for All-Inclusive Multi-Level Segmentation (AIMS), which segments images into three levels: part, entity, and relation. AIMS can also segment images based on mask prompts, which specify the region of interest. The paper proposes a unified AIMS model that uses multi-dataset multi-task training, task complementarity, association, and mask prompt encoder modules to address the challenges of annotation inconsistency and task correlation. The paper shows that the proposed method achieves better performance and generalization than existing methods on various segmentation tasks and datasets.

**Strengths:**

1. The AIMS task setup is interesting, which also includes the deduction of relationships between instances.
2. The prompt-based segmentation also makes sense, and it solves the multi-dataset training problem to some extent.
3. The proposed method achieves SOTA performance.

**Weaknesses:**

1. From the initial description of the proposed new task, it was difficult to understand why it would make sense to segment visual regions in relation level. Please clarify it.
2. The base framework consists of three separate decoders with a task complementarity module, However, the framework is relatively simple and lacks task-specific designs.
3. Regarding the operation in Formula 5, does G_split mean decompose the embedded feature into two features? The meaning of the symbol should be clearly described.
4. In Line 131, there is a mistake. "L_ce denote binary cross-entropy loss", It should be L_bce
5. The ablations in In Table 3 is incomplete. Compared to the baseline with three separate decoders for three levels, the addition of the association module improve the association performance and has some negative impact on segmentation performance on PPP. To verify the effectiveness of association module, I suggest the authors to experiment with both MPE and TCM modules and report the results.
6. It is not clear how entity-part and relation-entity association affect the final result separately. This should be included in the ablations.


**Questions:**

Please refer to weakness.

**Limitations:**

Limitations such as failure cases and border impact should be discussed.

---

> ### Author Rebuttal · Authors · 2023-08-09
>
> We thank the reviewer for the insightful comments to improve our paper further.
>
> ### Q1: Why it would make sense to segment visual regions in relation level?
>
> We introduce relation-level segmentation results in entity pairs for two key reasons.
>
> First, entity pairs represent the minimal relational constructs that can depict the entire scene graph of an image. This representation facilitates downstream tasks associated with scene graph analysis or generalized referring expression segmentation. As demonstrated in Table 5(a) of our paper and the subsequent table, the AIMS model contributes to enhanced performance, outperforming state-of-the-art methods in both panoptic scene graph (COCOPSG dataset) and generalized referring expression segmentation task (gRefCOCO) [1].
>
> | method | cIoU | gIoU |
> | :-----: | :----: | :----: |
> | ReLA [1] | 52.26 | 54.44|
> | Ours | 52.83 | 54.96|
>
> Second, the ability of our model to learn relation-level data allows it to learn richer and higher-level contextual features that can be used to discern visual entities. For example, given an image of a person sitting on a chair, the model can better segment the person and chair into two entity-level masks, if the model understands that some relation associates the person and chair. Although it is a relation-level subtask, it is beneficial to entity-level segmentation.  Our proposed task complementarity module can effectively propagate such useful relation-level information to the entity-level decoder. Furthermore, entity pairs are used to construct “pseudo” relation-level masks that serve as a form of mask prompt for our model to learn to split such relation-level masks into individual entity-level masks. This improves our model’s capability to split or subdivide images into masks. Table 1, 2 and 3 of our supplementary file show the benefits to the entity-level results by introducing the relation-level task.
>
> [1] GRES: Generalized Referring Expression Segmentation. CVPR 2023.
>
>
> ---
> ### Q2: The relative simplicity of the task complementarity module.
>
> Our baseline approach employs three distinct decoders for segmenting different levels, allowing each decoder to develop task-specific features for its particular task. The effectiveness of this separation, as opposed to using shared decoders, is substantiated in the comparison provided in the subsequent table. This comparison illustrates the advantages of our segmented approach, highlighting the strong need for separate network parameters for learning level-specific features. The values reported in PPP dataset are $AP^P$, $AP^E$ and $AR^{ER}$. The values reported in COCO-PSG dataset are $AP^R$, $AP^E$ and $AR^{RE}$.
>
> | Decoder Design | PPP | COCO-PSG |
> | :-----: | :----: | :----: |
> | Shared | 23.2/52.0/67.9 | 37.2/39.1/49.2 |
> | Separated | 24.5/53.4/69.7 | 38.9/40.4/50.9 |
>
> Building on the foundation of separated task-specific decoders, we introduce the task complementarity module to enable the sharing of each task's specific features with one another. Although this module is relatively simple, it proves efficient for multi-level segmentation in the Table 3(a) of our paper. Moreover, its streamlined design enhances adaptability across various base frameworks. For example, we successfully incorporate the task complementarity module into a Mask DINO by replicating the decoders three times for three-level segmentation (which serves as our baseline). The integration of this module in Mask DINO [2] follows a pattern similar to its implementation in Mask2Former, thereby showcasing the flexibility and wide applicability of our design. The values reported in PPP dataset are $AP^P$, $AP^E$ and $AR^{ER}$. The values reported in COCO-PSG dataset are $AP^R$, $AP^E$ and $AR^{RE}$.
>
> | Method | PPP | COCO-PSG |
> | ----- | :----: | :----: |
> | Mask DINO | 23.9/52.7/68.9 | 38.0/39.2/48.7 |
> | Mask DINO with TCM | 25.1/54.3/70.1 | 38.8/40.2/49.3 |
>
> [2] Mask dino: Towards a unified transformer-based framework for object detection and segmentation. CVPR 2023.
>
>
> ---
> ### Q3: Clarity issue on Formula 5.
>
> The $G_{split}$ means splitting the tensor along the channel dimension, which is the opposite of the concatenation operation.
>
>
> ---
> ### Q4: The modification from L_ce to L_bce.
>
> Thanks for pointing out this mistake. We will correct it in our paper.
>
>
> ---
> ### Q5: The ablation study on the baseline with TCM and MPE in Table 3(a)
>
> The table below presents an ablation study of the association module. It can be observed that the association module primarily enhances the association performance, having only a positive impact on the cross-level (entity-part/EP and relation-entity/RE) results. This finding aligns with the observations made in Table 3(a), where only slight variations in performance were noted. The values reported in PPP dataset are $AP^P$, $AP^E$ and $AR^{ER}$. The values reported in COCO-PSG dataset are $AP^R$, $AP^E$ and $AR^{RE}$.
>
> | Design | PPP | COCO-PSG |
> | ----- | ---- | ---- |
> | Baseline | 24.5/53.4/69.7 | 38.9/40.4/50.9|
> | Basline with TCM and MPE | 26.4/55.9/71.4 | 40.5/42.0/52.6|
> | Basline with TCM, MPE and AM | 26.5/56.1/72.3 | 40.5/42.1/53.1 |
>
>
> ---
> ### Q6: Independent influence of entity-part and relation-entity association.
>
> The table below illustrates the impact of two association-dependent modules on our final segmentation results. Notably, each association module exclusively influences the association performance between its corresponding two levels, without affecting the others. The values reported in PPP dataset are $AP^P$, $AP^E$ and $AR^{ER}$. The values reported in COCO-PSG dataset are $AP^R$, $AP^E$ and $AR^{RE}$.
>
> | Design | PPP | COCO-PSG |
> | ----- | ---- | ---- |
> | Baseline | 24.5/53.4/69.7 | 38.9/40.4/50.9|
> | Basline with EP association | 24.2/53.1/70.5 | 39.2/40.5/51.1 |
> | Basline with RE association | 24.4/53.3/69.9 | 39.0/40.5/52.0 |
> | Basline with EP and RE association | 24.3/53.1/70.6 | 39.0/40.5/52.0 |

---

> > ### Author Response · Authors · 2023-08-14
> > **Please let us know whether you have additional questions after reading our response**
> >
> > We appreciate your reviews and comments. We hope our responses address your concerns. Please let us know if you have further questions after reading our rebuttal.
> >
> > We hope to address all the potential issues during the discussion period.
> >
> > Thank you.

---

> > ### Author Response · Authors · 2023-08-16
> > **More clarification about limitations and border impact**
> >
> > ## Limitations
> >
> > Our work aims to segment an image into various regions across different hierarchical levels. Specifically, we define three explicit levels: relation, entity, and part, and utilize multiple datasets for supervised training. Consequently, the trained model may be biased toward human annotations. The categorization into these levels (especially part level) is highly subjective and might not always align perfectly with all downstream applications. The concurrent work, the SAM model, also suffers from this kind of problem and merely uses 3 tokens to learn segmentation at three levels implicitly. Class-agnostic segmentation can somehow alleviate this problem but cannot fully solve it. Exploring ways to segment images hierarchically in an unsupervised manner may present a more inspiring direction, potentially eliminating annotation and human biases. This is a promising direction that we leave as future work.
> >
> > ## Border Impact
> >
> > Our work can be used to obtain high-quality masks at multiple hierarchical levels. This high-quality mask can provide a smooth and minimal-effort experience for image editing users including amateurs and beginners. People who have image editing needs usually perform editing on the visual regions at the 3 hierarchical levels our model can handle. Our model can tremendously reduce the time, manual labor, and expertise required for selecting regions of interest for advanced image editing. On the other hand, this work will potentially bring a negative impact on the jobs and businesses of image editing experts, due to the lowered barrier to entry for amateurs who can easily perform advanced image editing with the help of our model’s high-quality hierarchical masks.
> >
> > ### We will add those parts to our paper. Thanks for your suggestions.

---

> > > ### Comment · Reviewer_Tt8E · 2023-08-16
> > >
> > > Thanks for your response. The rebuttal has addressed most of my concerns. I will raise my rating to 6.

---

### Official Review · Reviewer_u59B · 2023-07-01

**Soundness:** 3 good
**Presentation:** 2 fair
**Contribution:** 3 good
**Rating:** 6
**Confidence:** 4

**Summary:**

This work presents AIMS, a multi-level image segmentation model with levels representing parts, instance, and relation. Further, a curated dataset is created from several existing segmentation datasets. AIMS outperforms the baselines on the curated dataset.

**Strengths:**

1. The proposed architecture reasonably bridges information across levels.
2. The quantitative and qualitative performance are good.
3. Experiments, especially the ablations, are comprehensive.

**Weaknesses:**

1. It's unclear how the prompt mask is obtained and used during inference. When decoding with such a mask, would its experiments against baselines unfair comparisons since baselines might not have such additional localization information?
2. The comparison with SAM is partially done. Especially that SAM would provide fine segmentation masks with mask prompt with low-accuracy, while AIMS has not mentioned explicitly the quality of the mask prompt -- based on Figure 3 the masks seem to be accurate.
3. The presentation can be generally improved. Figure 2 provides details that distracts readers without further knowledge of more details, whilst the mask prompt encoder can be motivated earlier.


**Questions:**

1. Does the training set aggregation make any modification of the original datasets/annotations? Would the curated dataset (split) be released?
2. The comparison in Figure 4 only shows part segmentation. What would the comparison be when it comes to multi-instance level?
3. In table 5(b), shouldn't the $AR^E$ score for EntitySeg be bold since 87.2 > 87.1?

**Limitations:**

Limitations are not discussed by the authors. Apart from potential limitations in Weaknesses, I see there are these limitations:
1. The computational cost is unspecified.
2. The complementarity module and association module are designed to couple with the base framework, thus not obviously usable by other architectures.

---

> ### Author Rebuttal · Authors · 2023-08-09
>
> We thank the reviewer for the positive feedback and insightful comments.
>
> ### Q1: Unclear about the prompt mask used in inference.
>
> For a fair comparison, we conduct all three levels of segmentation using the full-image mask prompts throughout our experiments, without unfairly introducing any specific localization or region information. In Table 5(b), we show the entity- and part-level results between the SAM model and our method, employing only full-image mask prompts for the evaluation.
>
> To further substantiate the efficacy of our approach, we present a comparative analysis of the results in terms of AR@100. This comparison is conducted by prompting both SAM and our AIMS models with the same ground-truth entity mask for part-level inference. The outcomes, detailed in the following table, serve as additional evidence of the strengths of our method.
>
> | Method | PACO | ADE20K|
> | :-----: | :----: | :----: |
> | SAM | 54.4 | 50.6 |
> | Ours | 55.0 | 51.3 |
>
> ---
> ### Q2: The ablation study on the quality of mask prompt.
>
> In the following two tables, we show the ablation studies about the quality of mask prompt on PACO dataset where we degrade the ground truth entity masks with different levels including the part and entity level respectively.
>
> The performance comparison of part-level results is as follows:
> | Method | Ground-Truth Mask| Bounding Box | Randomly Extended Box | Randomly Created Mask |
> | ----- | ---- | ---- | ---- | ---- |
> | SAM | 54.4 | 53.5 | 52.8 | 52.5 |
> | Ours | 55.0 | 54.2 | 53.7 | 53.4 |
>
> The performance comparison of entity-level results is as follows:
> | Method | Ground-Truth Mask| Bounding Box | Randomly Extended Box | Randomly Created Mask |
> | ----- | ---- | ---- | ---- | ---- |
> | SAM | 100.0 | 98.7 | 98.6 | 98.9 |
> | Ours | 100.0 | 99.1 | 98.9 | 98.8 |
>
> For the extended bounding box, we randomly perturbed the four corners of the original bounding boxes. For the arbitrary mask, we randomly added polygon points to the extended bounding boxes to create random masks. We can provide the corresponding code to generate these boxes and masks in the next round discussion due to the limited characters.
>
>
> ---
> ### Q3: The improved presentation in Figure2.
>
> We have improved Figure2 with a better presentation. Please check the figure 1 and 2 in our submitted PDF.
>
>
> ---
> ### Q4: Training Set.
>
> We report about it in our supplementary file. All models are trained using five datasets: COCO, EntitySeg, PascalVOC Part, PACO, and COCO-PSG. Given that these datasets' original training and validation splits are tailored for single tasks, we collate the images and reorganize them to suit our AIMS task. Initially, we select 1069 and 1000 validation images from PPP (which covers the part and entity levels) and COCO-PSG (which covers the entity and relation levels), respectively. Following this, we eliminate any duplicate images in the unified training set that are present in the validation images, resulting in a refined training set comprised of approximately 236.7K unique images. In a nutshell, we only reorganized existing datasets and did not introduce new annotations.
>
> We will release the curated dataset to the community for future research and reproducibility.
>
>
> ---
> ### Q5: The wrong bold location.
>
> Thanks for pointing this out. We will correct it in our paper.
>
>
> ---
> ### Q6: Computation cost.
>
> The table below provides the parameter size of our largest model with the Swin-Large backbone. Notably, the new proposed modules, including the task complementarity module (TCM), mask encoder prompt (ME), and association module (AE), only introduce a minor increase of parameters compared to our baseline model (backbone and baseline decoder). This is due to the efficient shared weight design implemented in the three proposed modules. Overall, the number of parameters in AIMS with the Swin-Large backbone is 246,236,372 (200845932+42626304+2630144+1384+132608), compared to the 641,090,864 number of parameters in SAM with VIT huge backbone.
>
> | Backbone | Baseline Decoder  | TCM | ME | AM  |
> | :-----:  | :----:  | :----:  | :-----:  | :----:  |
> | 200,845,932 | 42,626,304 | 2,630,144 | 1,384 | 132,608 |
>
> For the computation cost, conducting full image inference under (800, 1333) image size in three levels requires 854.0 GFlops and takes approximately 0.142 seconds per image on A100 40G. This computation cost is comparable to the baseline method, which consumes about 801.0 GFlops and has an inference time of 0.138 seconds per image. For the SAM model we compare, it would cost 0.253 seconds per image within the same setting.
>
>
> ---
> ### Q7: The universality of proposed complementarity and association module.
>
> We introduce the task complementarity and association modules, drawing inspiration from the state-of-the-art segmentation decoder design, which employs learnable queries and transformer blocks. This approach allows for adaptability with minimal modification, aligning with designs commonly found in recent transformer-based segmentation methods [1,2,3]. For instance, we integrate these two modules into the improved Mask DINO [1] by replicating the decoders three times for three-level segmentation (our baseline). The incorporation of these modules in Mask DINO is carried out in a manner akin to their implementation in Mask2Former, demonstrating the versatility of our design.
>
> | Method | PPP  | COCO-PSG  |
> | :-----:  | :----:  | :----:  |
> | Mask DINO  | 23.9/52.7/68.9 | 38.0/39.2/48.7 |
> | Mask DINO with TCM and AM | 25.3/54.2/71.5 | 38.9/40.3/49.9 |
>
> [1] Mask dino: Towards a unified transformer-based framework for object detection and segmentation. CVPR 2023.
>
> [2] MaX-DeepLab: End-to-End Panoptic Segmentation With Mask Transformers. CVPR 2021.
>
> [3] k-means Mask Transformer. ECCV 2022.

---

> ### Author Response · Authors · 2023-08-14
> **Please let us know whether you have additional questions after reading our response**
>
> We appreciate your reviews and comments. We hope our responses address your concerns. Please let us know if you have further questions after reading our rebuttal.
>
> We hope to address all the potential issues during the discussion period.
>
> Thank you.

---

> > ### Comment · Reviewer_u59B · 2023-08-16
> >
> > I thank the authors for the informative rebuttal. They address most of my concerns and I'd like to raise my score to 6, under the assumption that some of the discussions would be included to further enhance the paper.

---

### Official Review · Reviewer_b8VN · 2023-07-06

**Soundness:** 3 good
**Presentation:** 3 good
**Contribution:** 3 good
**Rating:** 7
**Confidence:** 4

**Summary:**

This paper introduces a novel task called All-Inclusive Multi-Level Segmentation (AIMS) and proposes a unified AIMS model to address the challenges of annotation inconsistency and task correlation. The model consists of a shared image encoder and three independent decoders for part, entity, and relation predictions. It incorporates a Task Complementarity Module (TCM) to fuse task information and an Association Module to establish associations between different levels of segmentation. The model is trained using a **combination** of existing segmentation datasets. The model also incorporates a Mask Prompt Encoder (MPE) to provide supervision signals.

The proposed method outperforms SOTA methods on both panoptic part segmentation and panoptic scene graph estimation tasks. The proposed method also outperforms the Segment Anything model (SAM) on entity-level segmentation, even though the proposed method was trained on much less data than SAM.

**Strengths:**

### Novel approach to a multi-level segmentation task
* The paper introduces a novel task called All-Inclusive Multi-Level Segmentation (AIMS) and proposes a unified model to address the challenges of annotation inconsistency and task correlation. This task formulation and the proposed model are unique and provide a fresh perspective on multi-level segmentation.

### Comprehensive experiments with detailed ablations and good results
* The paper presents a comprehensive evaluation of the proposed method, comparing it with state-of-the-art methods and conducting ablation studies.
* The experiments are well-designed, and the results demonstrate the effectiveness and generalization capacity of the proposed method.
* The authors also provide detailed explanations of the model components and training settings, ensuring reproducibility.

I also appreciate the clarity of writing in this paper which has a thorough analysis of problems in existing segmentation frameworks motivating the need for the AIMS task.

**Weaknesses:**

### Missing performance analysis on images with a large number of objects
* SAM has shown impressive performance on images with a large number of objects. How does the proposed method fare in such conditions? It would be useful to break down the performance on the basis of the number of objects and compare it with SAM.

**Questions:**

* The authors mention that Swin-Large backbone provides the best results, but can you provide more details on the decoder architecture? Also, what is the total model size and computational requirement (FLOPs) of the model?
* The legend in Figure 2 is missing the symbol for "Add"

**Limitations:**

No limitations section.

---

> ### Author Rebuttal · Authors · 2023-08-09
>
> We thank the reviewer for the positive feedback and insightful comments.
>
> ### Q1: Missing performance analysis on images with a large number of obejcts.
>
> We present the recall@100 comparison of the entity-level results of between SAM and our model on selected LVIS validation data that includes more than 20 objects, as well as on crowd-human validation data. In the statistics, the average object numbers in our LVIS and crowd-human are 21 and 23, respectively. The table below demonstrates that our model can achieve performance comparable to the SAM model on images containing a large number of objects.
> | method | LVIS | crowd-human |
> | :-----: | :----: | :----: |
> | SAM | 37.2 | 97.5 |
> | Ours | 36.9 | 97.8 |
>
> Furthermore, we compare our method with the SAM model on the object clutter indoor segmentation for robotic grasping (OCID) and the few-shot segmentation (FSS) dataset on recall@100. These comparisons highlight some advantages of our method's separated-decoder design. This is because the SAM model does not explicitly divide the results into three levels and only selects the minimum loss corresponding to the ground truth for training. This approach leads to strong requirements for user selection effort at certain levels, whereas our design mitigates this limitation.
>
> | method | OCID | FSS |
> | :-----: | :----: | :----: |
> | SAM | 55.17 | 73.60 |
> | Ours | 76.63 | 86.26 |
>
>
> ---
> ### Q2: The details of decoder structure.
>
> We adhere to the transformer decoder design in the Mask2Former method, encompassing a set of learnable queries and nine transformer blocks. Each block comprises a cross-attention layer, a self-attention layer, and a feed-forward network. In our AIMS decoder, we begin with three distinct transformer decoders designated for different levels of segmentation results, forming our baseline approach.
>
> Building upon this baseline, we introduce the task complementarity module to facilitate information fusion across different levels, such as the fusion between entity-part and relation-entity decoders. This innovative design is inspired by our observation that separate transformer decoders consistently outperform a shared transformer decoder; such observation is demonstrated in the following table. Essentially, our experimental findings support the notion that decoders at different levels possess distinct unique features compared to other decoders. The values reported in PPP dataset are $AP^P$, $AP^E$ and $AR^{ER}$. The values reported in COCO-PSG dataset are $AP^R$, $AP^E$ and $AR^{RE}$.
>
> | Decoder Design | PPP | COCO-PSG |
> | :-----: | :----: | :----: |
> | Shared | 23.2/52.0/67.9 | 37.2/39.1/49.2 |
> | Separated | 24.5/53.4/69.7 | 38.9/40.4/50.9 |
>
> Our proposed task complementarity module capitalizes on these individual strengths, enabling each level decoder to interact with and be enriched by the others. This synergistic approach improves performance compared to our baseline, highlighting the efficacy of tailored, level-specific interactions within the decoder architecture.
>
> For the association module, we directly sample positive embeddings from each level generated by the final transformer block to establish associations between two adjacent levels. It's important to note that these sampled positive embeddings are pre-processed through a fully connected (FC) layer.
>
>
> ---
> ### Q3: Model size and computational requirement.
>
> The table below provides the parameter size of our largest model with the Swin-Large backbone. Notably, the new proposed modules, including task complementarity module (TCM), mask encoder prompt（ME）, and association module (AE), only introduce a minor increase of parameters compared to our baseline model (backbone and baseline decoder). This is due to the efficient shared weight design implemented in the three proposed modules. Overall, the number of parameters in AIMS with the Swin-Large backbone is 246,236,372 (200845932+42626304+2630144+1384+132608), compared to the 641,090,864 number of parameters in SAM with VIT huge backbone.
>
> | Backbone | Baseline Decoder  | TCM | ME | AM  |
> | :----:  | :----:  | :----:  | :-----:  | :----:  |
> | 200,845,932 | 42,626,304 | 2,630,144 | 1,384 | 132,608 |
>
> For the computation cost, conducting full image inference under (800, 1333) image size in three levels requires 854.0 GFlops and takes approximately 0.142 seconds per image on A100 40G. This computation cost is comparable to the baseline method, which consumes about 801.0 GFlops and has an inference time of 0.138 seconds per image. For the SAM model we compare, it would cost 0.253 seconds per image within the same setting.

---

> > ### Comment · Reviewer_b8VN · 2023-08-13
> >
> > I appreciate the authors' responses. They have answered all my questions. I think this paper makes a good contribution and would like to maintain my rating as "Accept".

---

### Official Review · Reviewer_mZLg · 2023-07-10

**Soundness:** 3 good
**Presentation:** 3 good
**Contribution:** 3 good
**Rating:** 6
**Confidence:** 5

**Summary:**

This work proposes a unified multi-level segmentation (AIMS) approach. For better generalisation, the model is concurrently trained on multiple dataset consisting varied hierarchical level annotations across parts, entities and relations. In order to utilise signals from multiple hierarchical level as well as infuse model with level-awareness for each training sample, AIMS proposes three modules - task complementarity, association and mask prompt encoder. Experimental evaluation shows AIMS performing better than other state-of-the-art class-aware as well as class-agnostic segmentation.

**Strengths:**

Originality
Although model is inspired from Mask2former, the inclusion of other novel modules packages AIMS for multi-level hierarchical segmentation. Thus I note sufficient novelty in the proposed architecture.

Quality
The paper is well-written and easy to follow. Authors can be more clearer upfront about AIMS being complete class-agnostic model (i.e., mask proposal model) and there is no semantic labelling.

Significance
1. As far as I am aware of, AIMS is the first segmentation model to be trained with multi level semantic understanding. Though SAM is trained with fine-grained annotations it lacks part- and relation-level understanding and thus certain fine-grained semantic features may not be fully segmented out.
2. Moreover, training with multiple datasets enables AIMS to be deployed for range of image editing and manipulation applications.
3. AIMS achieves these advantages with as minimal data as possible (237K vs 11M in SAM).

**Weaknesses:**

1. As pointed earlier, AIMS is class-agnostic. Hence it cannot label the output segments.
2. AIMS use interactive settings by deploying mask prompt encoder (MPE) to specify model what to segment. However, MPE usage is limited to providing unmask region of an input image for model to act on. There is no provision to use text, bounding box, or points as inputs. This is very limited interactive settings.
3. For the same reason though powerful, AIMS cannot extract referring expression segmentation. This is despite the fact that AIMS is trained with semantic of relation understanding.
4. Overall, the model is novel but the evaluation lacks distinguishing aspects that AIMS in the very first place was designed for.

**Questions:**

AIMS model have three separate output while other models output single mask irrespective of entity our parts. So in table 5(b), how do one measure seperate AR for parts and entity segmentation. Do you run seperate inference with different prompt for part level and entity level ? Or full-image mask prompt is applied ?

**Limitations:**

Authors haven't addressed any limitations. Some of the listed weaknesses above can be elaborated to address the limitations the paper.

---

> ### Author Rebuttal · Authors · 2023-08-09
>
> We thank the reviewer for the positive feedback and insightful comments.
>
> ### Q1: AIMS is class-agnostic model and cannot label the output segments.
>
> Thanks for the suggestion. Although mask labeling is related to our work, the main goal of our work is to develop an all-inclusive segmentation model capable of delivering high-quality multi-level mask proposals for various downstream tasks, including class-aware ones. That's because class-agnostic segmentation has shown great effectiveness to the unseen categories or image domain than class-aware one in entity segmentation [1,2] or SAM [3]. In Table 5(a) of our paper, we demonstrate the advantages of fine-tuning the AIMS pre-trained model for class-aware tasks such as panoptic part segmentation and scene graph. This approach leads to performance that outperforms the state-of-the-art methods of each task, showcasing the flexibility and effectiveness of our model even on class-aware tasks.
>
> Furthermore, AIMS works complementarily with existing mask labeling methods [4,5,6] that can be applied to external mask proposals.
>
> [1] Open World Entity Segmentation. TPAMI 2022.
>
> [2] High-Quality Entity Segmentation. ICCV 2023.
>
> [3] Segment Anything. ICCV 2023.
>
> [4] Open-vocabulary semantic segmentation with mask-adapted clip
> . CVPR 2023.
>
> [5] Open-Vocabulary Universal Image Segmentation with MaskCLIP. ICML 2023.
>
> [6] Scaling Open-Vocabulary Image Segmentation with Image-Level Labels
> . ECCV 2022.
>
>
> ---
> ### Q2: Limited interactive settings like text, bounding box or points.
>
> Our AIMS model is not designed for interactive segmentation. Instead, we focus on automatic image segmentation at multiple explicit levels in an all-inclusive manner, without requiring intricate prompt engineering and parameter tuning during inference to obtain desired results, as SAM needs. SAM's weakness is that it does not enforce explicit levels of segmentation and only selects the prediction that has the smallest loss against ground truth for training. Therefore, our setting and the proposed method is more desirable for large-scale automated segmentation mask generation without costly human intervention and correction.
>
> Interactive settings are usually sensitive to geometric prompts. For instance, using two-point prompts with slightly different locations in SAM can give very different mask predictions. Furthermore, for many applications that do not rely on user inputs (e.g., automated image content analysis), using a regular grid of points as prompts is unfavorable since the model can fail to segment objects that do not overlap with those points.
>
> While our proposed framework can accept mask prompts, this feature is part of our design to address annotation inconsistencies among datasets. It encourages the network to learn how to perform further separation on a given mask instead of functioning as an interactive segmentation component. We do not expect users to be able to draw out sufficiently good mask prompts for our model.
>
>
> ---
> ### Q3: Lacking ability for referring expression segmentation.
>
> Similar to the first question, one of AIMS's roles is to obtain good pretrained weights for finetuning on downstream tasks. We test AIMS model in generalized referring expression segmentation [7] on gRefCOCO dataset.
>
> The detailed structure design proceeds as follows: First, we obtain the text embedding using the BERT model. Within each level of the decoder, we integrate three additional transformer blocks. These blocks include a cross-attention layer between the query and text embedding, a cross-attention layer between the query embedding and image features, self-attention, and a feedforward network. For each cross-attention, query embedding is the query and other keys and values. These additional layers are appended to the default nine transformer layers. Ultimately, the three-level results are merged for bi-partite matching. During inference, we directly select the mask with the highest score. To ensure a fair comparison, we employ a similar number of training iterations to the ReLA method [7], utilizing the Swin-Tiny backbone.
>
> The comparison results are presented in the table below, demonstrating the generalization ability of the AIMS model in the context of referring expression segmentation.
>
> | method | cIoU | gIoU |
> | :-----: | :----: | :----: |
> | ReLA [7] | 52.26 | 54.44|
> | Ours | 52.83 | 54.96|
>
> [7] GRES: Generalized Referring Expression Segmentation. CVPR 2023.
>
>
> ---
> ### Q4: Evaluation Metrics.
>
> Our evaluation metrics face the complex challenge of assessing three performance levels simultaneously, making it difficult to devise a unified metric that accurately balances the weight across these levels. Consequently, we opt for decoupled evaluation metrics tailored to each level, ensuring a better model demonstrates enhanced performance across all three levels. This approach not only allows us to pinpoint specific areas of performance improvement when implementing new designs but also helps validate our method's effectiveness.
>
> For unified evaluation metrics, we explore it in downstream tasks such as PartPQ and PWQ for panoptic part segmentation and R/mR@100 for panoptic scene graph construction, as shown in Table 5(a) of our paper. That shows our model also performs better than state-of-the-art methods by using unified evaluation metrics.
>
> ---
> ### Q5: The detailed inference setting in Table 5(b).
> Using a full-image mask prompt, we compare the entity- and part-level results between SAM and our model.  To ensure a fair comparison in the part-level results, we show the result comparison in AR\@100 in the following table by utilizing the ground-truth entity-level masks as mask prompts for part-level inference.
>
> | method | PACO | ADE20K|
> | :-----: | :----: | :----: |
> | SAM [3] | 54.4 | 50.6 |
> | Ours | 55.0 | 51.3 |

---

> ### Author Response · Authors · 2023-08-14
> **Please let us know whether you have additional questions after reading our response**
>
> We appreciate your reviews and comments. We hope our responses address your concerns. Please let us know if you have further questions after reading our rebuttal.
>
> We hope to address all the potential issues during the discussion period.
>
> Thank you.

---

### Author Rebuttal · Authors · 2023-08-09


We thank the reviewers for the insightful comments regarding our work. We have carefully addressed each of your concerns, and our responses can be found in the respective rebuttal sections for each reviewer.


In addition, we have included a PDF containing three figures, as requested by some of the reviewers. We will refine our paper in light of your insightful advice.

---

### Decision · Program_Chairs · 2023-09-21

**Decision:**

Accept (spotlight)

**Comment:**

This submission received five reviews all of which are positive and propose acceptance. The authors address technical questions in the discussion to the satisfaction of the reviewers and some increased the scores.

All reviewers acknowledge this paper as novel and they find the idea of combining a segmentation to include different tasks, especially also semantic relationship as interesting. The experiments are reported and appreciated as being better than current SOTA, comparison with current best performing models are included, (e.g. SAM) and the segmentation results come out better.
In the weakness section of the reviews are mostly technical points that have been explained in the rebuttal. There is no major shortcoming that would stand in the way of accepting this version already.

With five votes for acceptance this is a clear contribution to the field and a solid paper.